# Temozolomide Treatment Increases Fatty Acid Uptake in Glioblastoma Stem Cells

**DOI:** 10.3390/cancers12113126

**Published:** 2020-10-26

**Authors:** Seamus Caragher, Jason Miska, Jack Shireman, Cheol H. Park, Megan Muroski, Maciej S. Lesniak, Atique U. Ahmed

**Affiliations:** Department of Neurological Surgery, Feinberg School of Medicine, Northwestern University, 303 E Superior Street, Chicago, IL 60611, USA; seamus_caragher@hms.harvard.edu (S.C.); jason.miska@northwestern.edu (J.M.); Jshireman@wisc.edu (J.S.); cheol.park@northwestern.edu (C.H.P.); Megan.muroski@milliporesigma.com (M.M.); maciej.lesniak@northwestern.edu (M.S.L.)

**Keywords:** glioblastoma, glioma stem cell, metabolism, therapeutic stress, chemoresistance

## Abstract

**Simple Summary:**

Patients diagnosed with glioblastoma (GBM) brain tumors typically survive less than two years, despite aggressive therapy with surgery, radiation, and chemotherapy. A major factor underlying this lethality is the ability of GBM tumors to adapt to stress, including the stress of treatment. The role of metabolism in this process remains incompletely understood. We, therefore, explored the connection between cellular phenotype, chemotherapeutic stress, and metabolism in GBM. We found that inducing changes in GBM phenotypes led to alterations in metabolic behavior. Further, during treatment with chemotherapy, GBM cells that became resistant to therapy increased their fatty acid uptake. These therapy-induced alterations in nutrient uptake may underlie therapy resistance and deadly recurrence.

**Abstract:**

Among all cancers, glioblastoma (GBM) remains one of the least treatable. One key factor in this resistance is a subpopulation of tumor cells termed glioma stem cells (GSCs). These cells are highly resistant to current treatment modalities, possess marked self-renewal capacity, and are considered key drivers of tumor recurrence. Further complicating an understanding of GBM, evidence shows that the GSC population is not a pre-ordained and static group of cells but also includes previously differentiated GBM cells that have attained a GSC state secondary to environmental cues. The metabolic behavior of GBM cells undergoing plasticity remains incompletely understood. To that end, we probed the connection between GSCs, environmental cues, and metabolism. Using patient-derived xenograft cells, mouse models, transcriptomics, and metabolic analyses, we found that cell state changes are accompanied by sharp changes in metabolic phenotype. Further, treatment with temozolomide, the current standard of care drug for GBM, altered the metabolism of GBM cells and increased fatty acid uptake both in vitro and in vivo in the plasticity driven GSC population. These results indicate that temozolomide-induced changes in cell state are accompanied by metabolic shifts—a potentially novel target for enhancing the effectiveness of current treatment modalities.

## 1. Introduction

Glioblastoma (GBM), the most aggressive and prevalent primary brain tumor afflicting adults, remains intractable despite decades of research [1]. One theory as to why GBM tumors resist both broad acting and targeted intervention is the marked heterogeneity of GBM tumors [2,3,4]. Indeed, it has previously been shown that GBM tumors are unique down to the single-cell level and are made up of a wide variety of cellular subpopulations, including glioma stem cells (GSCs) [2,5]. These cells are characterized by the ability to withstand therapy and endowed with heightened self-renewal potential [6,7]. This ability of GSCs to resist current therapies, including radiation and chemotherapy, is thought to play a critical role in GBM recurrence. Further complicating this picture, we and others have shown that GBM cells, rather than existing as a static and unchanging phenotype, exhibit a high degree of cellular plasticity—the ability to interconvert between multiple cellular states in response to microenvironmental cues such as hypoxia [8,9,10]. Our group has shown that treatment with standard of care temozolomide (TMZ) drives alterations in GBM cells and pushes them towards the GSC state [11]. Other groups have shown similar results following radiation treatment [12]. These two facets of GBM—wide-ranging heterogeneity and interconversion of individuals cells between different phenotypes—underlie the stark lack of success in treating GBM. These tumors are diverse and able to react to therapeutic intervention rapidly.

In light of this complex interplay between environment, phenotype, and tumor progression, understanding the mechanisms that underlie this plasticity is critical. To that end, a range of phenotypic domains has been examined for contributions to tumor plasticity during therapy, including epigenetic modification and gene expression [8,9]. Recently, the metabolic phenotype of GBM tumors has emerged as an exciting area of research (expertly-reviewed [13,14]). GBM tumors have been shown to rely on glycolysis during phases of growth [15], with inhibition of glycolysis, improving the survival of mouse models of GBM [16,17]. Other groups have highlighted the importance of fatty acid oxidation in GBM growth [18]. Critically, metabolism in brain tumors is heterogeneous and linked to stemness. For example, a GSC population has been demonstrated to exhibit elevated lipid levels and utilize fatty acids to withstand nutrient deprivation [19]. Other reports have highlighted the importance of fatty acid synthesis in enabling the adoption of a GSC state [20].

This link between metabolism, tumor heterogeneity, and plasticity raises the question of how these forces interact during times of stress. Recent work has shown that GSC populations in brain tumors rely on enhanced glucose uptake and resultant glycolytic metabolism to thrive under metabolic stress [21,22]. A recent study showed that GSCs could exhibit distinct metabolic heterogeneity, which can respond to stress by switching metabolic phenotype [23]. This work was further corroborated by a recent report showing alterations in fatty acid metabolites can drive adaptive growth via paracrine signaling [24]. These exciting reports raise the question of how chemotherapeutic stress specifically alters GBM metabolic behavior, plasticity, and therapy resistance. We sought to probe this question and hypothesized that the standard of care TMZ treatment would alter the metabolic behavior of GBM cells in vitro and in vivo.

## 2. Results

### 2.1. Culture Conditions and Cell State Alter Metabolic Phenotypes In Vivo

To begin our investigation into the connection between cellular phenotype and metabolic phenotype, we first compared the metabolic activity of patient-derived xenografts (PDX) GBM 43 cells in culture conditions known to induce different cellular phenotypes. GBM cells are known to be highly sensitive to culture conditions [25,26]. GBM43 cells were cultured either as neurospheres in media that promotes a GSC fate (Neurobasal supplemented with N2, bFGF, EGF [27]) or compared to adherent in conditions known to force differentiation (1% fetal bovine serum) [28]. After eight days in culture, cells were collected, and a mitochondrial stress test was performed utilizing the Seahorse Extracellular Flux analysis, a well-established method for assessing mitochondrial respiration and glycolysis [29]. Briefly, cells were cultured in a limited substrate media, adhered to microplates at equal cell densities, and oxygen consumption rate measured while specific electron transport chain inhibitors [29] were added to the media to characterize the mitochondrial phenotype of the cells. We found that simply culturing PDX GBM cells in stem cell promoting media decreased their metabolic activity, including basal oxygen consumption rates (OCR), maximal OCR, and their spare respiratory capacity relative to differentiated GBM cells (Figure 1A,B: CSC culture vs. Differentiated culture conditions: Basal OCR *p* < 0.001, Maximal OCR *p* < 0.001, Spare Respiratory Capacity *p* < 0.01). These data suggest that alterations in the cell phenotype, in response to microenvironmental cues, also induce marked changes in the metabolic behavior in GBM cells, in line with previously published results [30].

Given this observed difference in metabolism between GSCs and differentiated GBM cells, we next sought to identify the particular metabolic process at play in each group. Two well-established pathways for energy generation rely on the uptake and breakdown of glucose and fatty acids [31]. In order to determine the contribution of fatty acid metabolism to this observed connection between cell state and metabolism, we performed extracellular flux analysis on cells cultured in CSC or differentiation media, in combination with compounds to interrogate fatty acid metabolism. Following 8 days culture, the analysis was performed on cells from both culture conditions in the presence of Bovine Serum Albumin (BSA, a control), etomoxir (Eto, an inhibitor of palmitate uptake into the mitochondria), 150 µm palmitate conjugated BSA (Palm, a fatty acid utilized for cellular respiration), or a combination of Eto and Palm. Comparisons among these groups enabled us to determine the relative dependence on cells of exogenous and endogenous fatty acids. This analysis showed that GSCs rely on both types of fatty acids, whereas differentiated cells are more reliant on exogenous fatty acids (Figure 2A,B). Thus, it appears that the observed alterations in metabolic phenotype may depend on alteration in fatty acid metabolism and the ability of GSCs to increase fatty acid utilization.

Due to the conditions utilized in Figure 1 and Figure 2A,B, forcing cells to take on a certain phenotype, they do not necessarily reflect the behavior of cells existing in a heterogeneous tumor population. As the low number of GSCs and other technical issues precluded the isolation of sufficient GSCs for extracellular flux analysis, we decided to utilize flow-cytometry based metabolic assays to determine the connection between cell state and metabolism in heterogenous tumor populations. Using two PDX GBM lines, GBM 12 and GBM43, glucose uptake was determined using fluorescently tagged 2-NDBG, while fatty acid uptake was determined using our proprietary qDOT-conjugated palmitate [32]. GSCs were identified using anti-CD133 antibodies [33]. Analysis of both the fatty acid and glucose uptake within the GSC and non-GSC populations revealed preferential uptake of fatty acid by GSCs, with only minor differences in glucose uptake across the two populations (Figure 2C). These results further corroborate the extracellular flux assay and demonstrate a connection between the cellular and metabolic phenotypes of GBM cells. Further, they reinforce the notion that GSCs have a high reliance on fatty acids.

### 2.2. Chemotherapeutic Stress Alters Metabolic Phenotype

These data show that GSCs have preferential uptake of fatty acids and that induction of a GSC state increases this preference. We have shown previously that treatment with TMZ at physiologically relevant concentrations is sufficient to shift GBM cells to a GSC state [11]. Given these observations, we next investigated how chemotherapy influences metabolism in both differentiated cells and GSCs. To that end, PDX GBM 43 cells were treated with 50 μM TMZ for 96 h, and their metabolic state was assayed by extracellular flux analysis. This assay revealed that TMZ significantly enhances mitochondrial metabolism, including increased basal oxygen consumption, nearly threefold higher maximal oxygen consumption rate, and doubled spare respiratory capacity (Figure 3A: basal OCR *p* < 0.001, maximal OCR *p* < 0.001, spare respiratory capacity *p* < 0.01). Thus, it appears that TMZ treatment increases mitochondrial metabolism. In order to assess the stability of this altered metabolic phenotype, we next performed the same experiment on cells pulse treated with TMZ for 18 h and then left for either 6 or 8 days. This analysis showed that pulse treatment actually induced the adoption of metabolic quiescence (Appendix A), suggesting that heightened metabolism during acute therapy leads to a different phenotype following therapeutic intervention.

In light of the results suggesting that fatty acid uptake is critical for GSCs, we next re-did the TMZ assay in the presence of palmitate in order to determine how exogenous fatty acids influence this behavior. In addition, the experiment was also run with Etomoxir. The addition of fatty acids did not further enhance the metabolic shift associated with TMZ, suggesting that this increased metabolic capacity is linked to endogenous fatty acids (Figure 3B: basal OCR-TMZ + PALM *p* > 0.05 compared to PALM alone; maximal OCR-TMZ/TMZ + PALM *p* > 0.005 vs. CTRL and PALM). Critically, the addition of Eto to the experiment abolished this TMZ-induced increase in metabolic activity. Coupled together, these data indicate that TMZ treatment increases oxidation of endogenous fatty acids.

### 2.3. Microarray Analysis Reveals Chemotherapeutic Stress Increase Expression of Genes Related to Fatty Acid Oxidation

Given this increase in fatty acid metabolism, we next sought to determine the underlying mechanism. To that end, we performed a microarray analysis on PDX GBM43 cells treated with 50 μM TMZ for 8 days. Microarray analysis revealed that TMZ-treated cells preferentially upregulate a variety of genes related to fatty acid oxidation, as well as the citric acid cycle (Figure 3C and Appendix A), indicating that the metabolic plasticity observed during TMZ treatment occurs concurrently with robust alterations in the transcription of genes known to drive fatty acid uptake and use. In sum, these results show that TMZ treatment increases the avidity of GBM cells for fatty acids.

### 2.4. Chemotherapeutic Stress Shifts GBM Metabolism towards Lipid Metabolism

These results indicate that TMZ treatment alters metabolic behavior. In order to better characterize this process, we next examined the uptake of fatty acids and glucose during chemotherapeutic stress. Cells were treated with 50 μM TMZ or equimolar DMSO as vehicle control and analyzed following 2, 4, 6, or 8 days. We found that TMZ induced a time-dependent increase in fatty acid uptake. This increase culminated on day 8 when nearly all cells imported fatty acids. GSCs had extremely high basal uptake of fatty acids, consistent with our results from Figure 1; however, they also showed increased fatty acid uptake relative to same day DMSO controls. Glucose uptake was severely decreased in GBM cells following TMZ treatment (Figure 4 and Appendix A). Thus, these results indicate that TMZ strongly influences how GBM cells take in nutrients and induce a shift towards lipid metabolism.

Of note, GBM tumors are broken into several subtypes based on their genetic expression profiles. These subtypes include classical, proneural, and mesenchymal [34]. These subtypes are associated with different driver mutations, responses to therapy, and prognosis [35]. As these experiments were all performed in PDX GBM43 cells, which are proneural, we next sought to determine if this effect was cell line or subtype dependent. Thus, we repeated the same experiment in three other GBM cell lines: GBM12 (proneural), U251 (mesenchymal), and GBM6 (classical) (see Table 1 for genetic information). Remarkably, we observed the same increase in TMZ-dependent increase in all cell lines. However, the effects were most pronounced in the other proneural cells tested, GBM12. GBM6 and U251 showed significantly, but more moderate, increases in fatty acid uptake (Figure 5 and Appendix A). In sum, these results show that chemotherapeutic stress pushes GBM cells to adopt a metabolic phenotype associated with lipid uptake.

### 2.5. Temozolomide Alters Metabolic Behavior In Vivo

To this point, all data presented have been from cultured cells. In order to determine if these metabolic shifts are maintained in vivo, we implanted 150,000 PDX GBM43 cells into the brains of athymic nude mice (nu/nu). Following 7 days to allow for implantation, mice received either 2.5 mg/kg TMZ or equimolar DMSO (i.p.) each day for five consecutive days. Following 3 days, mice were sacrificed, and whole brains rapidly removed. Cells were then incubated with fluorescently tagged glucose and fatty acid analogs, as well as anti-CD133 and anti-HLA antibodies to identify GSCs and exclude mouse brain cells, respectively. Flow cytometry was performed as in previous experiments. Critically, we found that treatment with TMZ increases the uptake of fatty acids and decreases the uptake of glucose (Figure 6: DMSO vs. TMZ, *p* < 0.05). When examining the GSC and non-GSC compartments specifically, we found that the apparent increase in fatty acid uptake was found mainly within the CD133+ population, in accordance with our in vitro observations (Figure 6 and Appendix A). In contrast, the decreased glucose uptake was observed in the CD133− population, further illustrating GSC specific behaviors during chemotherapeutic stress.

## 3. Discussion

This study illustrates the complex interplay between cellular phenotypes, metabolic behavior, and chemotherapeutic stress. They demonstrated that altering GBM phenotype by forced differentiation altered metabolism, inducing quiescence. Further, GBM cells treated with standard of care chemotherapy shifted their metabolic behavior to uptake more fatty acids. Interestingly, this behavior was most pronounced in therapy induced GSCs.

The precise link between alkylation of the DNA backbone and lipid uptake, however, is not clear. Examination of the available literature suggests several reasons why therapeutic stress would induce increased fatty acid uptake and metabolism. First, it has been shown that cancer cells rely on fatty acids to provide the cellular components required rapid proliferation, including membrane phospholipids and signaling lipids (expertly reviewed by Currie et al. [36]). In this framework, our result would seem to suggest that GBM cells encountering therapeutic stress rapidly increase uptake of lipids to provide the necessary energy substrates to fuel cellular response processes. This fits with previously published reports that GBM cells utilize fatty acids under other forms of stress, including metabolic stress [13,24]. Second, DNA damage is well established to cause increases in the production of reactive oxygen species (ROS). Data relating fatty acids to the production of ROS in the mitochondria suggests that fatty acids can exert a bivalent effect, increasing ROS under certain conditions and reducing their generation under others [37]. Thus, it is not impossible that increased fatty acid uptake represents an attempt by GBM cells to counter TMZ-induced ROS generation.

One key question is how the subtype of specific cells influence this behavior. As we observed in our time course nutrient uptake studies, the subtype of the cell line in question influence the extent of the metabolic alterations. Specifically, it appears that proneural GBM cell lines, such as GBM43 and GBM12, exhibited the greatest alterations in metabolism following treatment with TMZ, while classical and mesenchymal subtypes showed less pronounced shifts in nutrient uptake. A recent paper provides critical insight into this subtype specific phenomenon. Marziali et al. [38] showed that GSC cell lines clustered by genetic expression fall roughly into two categories, proneural-like and mesenchymal-like. Critically, their analysis of metabolic phenotypes found that these two genetic clusters correspond with two distinct metabolic phenotypes—proneural-like samples exhibit lower lipid metabolism, while mesenchymal-like cells have higher lipid metabolism. Our data correlate with these data, with base line uptake of fatty acids higher in U251 mesenchymal cells than in either proneural GBM12 or GBM43. Further, recent research demonstrates the so-called “proneural-mesenchymal shift”, in which proneural cells subjected to therapeutic stress attain a mesenchymal status and induce recurrence [39]. Thus, our data demonstrating shift of proneural cells increase lipid uptake, and a metabolic status associated with mesenchymal tumors, following therapy might suggest metabolic changes correlated with or participate in this proneural-mesenchymal shift. Another potential explanation comes from recent work highlighting differential Notch pathway activation in altering metabolic phenotype among GSCs [40]. Further research will determine the extent to which our observed metabolic shifts correlate with other aspects of plasticity.

Given the well-known ability of TMZ to induce cellular conversion to the GSC state [11,41], we suspect that increased fatty acid by CD133 negative cells represents a functional shift towards GSC status that correlates with genetic and transcriptomic changes. Indeed, this general reliance of GSCs on fatty acids is in line with recent results demonstrating that cancer stem cells in pancreatic tumors are dependent on fatty acid and oxidative phosphorylation [42]; certainly, it has been shown in breast cancer that induction of fatty acid utilization drives self-renewal and chemo-resistance in breast cancer stem cells [43]. It remains an open question in GBM whether transition to the GSC state relies on fatty acid metabolism or if fatty acid metabolism is an indicator of the transition. Recently it was shown that dormant GSCs may exhibit a distinct metabolic phenotype that participates in their reaction to chemotherapy [44]. In addition, it has been shown that GSCs reacting to stress may alter their metabolic phenotype to avoid the production of ROS products [45]. Finally, Gimple et al. [46] recently demonstrated that GSCs increase fatty acid synthesis and utilization to drive EGFR signaling, further connecting metabolic plasticity and stemness. Finally, recent reports of TMZ resistant GBM cell lines have been shown to upregulated lipid metabolite secretion and alterations in mitochondrial dynamics [47]. These data are in line with our results illustrating increased fatty acid uptake during and following TMZ treatment linked to cellular plasticity. Coupled together, they further underlie the potential of fatty acid metabolism as a participant in TMZ resistance and GBM recurrence. Future research will continue to illuminate this interesting interaction between stemness and metabolic phenotypes.

This study also prompts the question of how similar this observed GSC preference for fatty acids is to embryonic or healthy stem cells. Available evidence suggests that in this preference for fatty acids, GSCs may be recapitulating the behavior of non-cancerous stem cells. For example, it has been shown in hematopoietic stem cells that fatty acid uptake and oxidation are critical for proper maintenance of the stem cell state; loss of key fatty acid oxidation enzymes lead to the loss of stem cell maintenance [48]. Further, neural stem cells, healthy counterparts to the GSCs investigated here, rely on lipid metabolism and lipid synthesis to drive proliferation [49]. Thus, GSC preference for fatty acids, accentuated during times of therapeutic stress, represents the co-opting of metabolic mechanisms for maintaining healthy stem cell populations.

This study is not without limitations. First, the technical difficulty of culturing a large number of GSCs limited our ability to probe a confirmed post-TMZ sample of GSCs by Seahorse. Second, GBM tumor cells are exquisitely sensitive to changes in culture conditions [25,50]. While we were able to confirm our major finding in a mouse model, there is always concern that findings may not perfectly recapitulate the physiological conditions of GBM tumors in human patients. Third, we only investigated a specific subtype of fatty acids. Lipid metabolism in cancer is an expansive area of research [51,52,53]; we hope that further exploration will reveal further details about the connection between specific lipid particles and GBM plasticity. Finally, these experiments were conducted in cell lines that are all isocitrate dehydrogenase (IDH1/2) wild type (see Section 4.1). IDH status is a key driver of differential prognosis in GBM [54,55]. As such, these data can only be applied to IDH wildtype cell lines and tumors. Further research will establish the role of IDH mutations in GBM therapy-induced plasticity. Despite these limitations, this study highlights the importance of metabolic plasticity in GBM and its potential as a therapeutic target for increasing the efficacy of TMZ.

## 4. Materials and Methods

### 4.1. Cell Culture

U251 human glioma cell line were procured from the American Type Culture Collection (ATCC; Manassas, VA, USA). These cells were cultured in Dulbecco’s Modified Eagle’s Medium (DMEM; HyClone, Thermo Fisher Scientific; San Jose, CA, USA) supplemented with 1% fetal bovine serum (Atlanta Biologicals, Lawrenceville, GA, USA) and 2% penicillin-streptomycin antibiotic mixture (Cellgro; Herdon, VA, USA; Mediatech, Herdon, VA, USA). Patient-derived xenograft (PDX) glioma specimens (GBM43, GBM12, GBM6, GBM5, and GBM39) were obtained from Dr. C. David James at Northwestern University, Chicago, IL, USA and maintained according to published protocols [56]. Of note, these cells represent a range of different tumor subtypes with unique combinations of genetic alterations [57].

Cells were propagated in vivo by injection into the flank of nu/nu athymic nude mice. In vitro experiments with these cells were performed utilizing DMEM supplemented with 1% FBS and 2% penicillin-streptomycin (P/S) antibiotic mixture. All cells were maintained in humidified atmosphere with CO_2_ and temperature carefully kept at 5% and 37 °C, respectively. Dissociations were performed enzymatically using 0.05% trypsin and 2.21 mM EDTA solution (Mediatech, Corning; Corning, NY, USA). For experiments, cells were cultured in their appropriate cell culture media treated with temozolomide diluted in DMSO (TMZ; Schering Plough; Kenilworhth, NJ, USA) or equimolar DMSO vehicle control. Cell lines for each experiment are as follows:
 Figure 1, Figure 2, Figure 3 and Figure 4: GBM43 (proneural) Figure 5: GBM43, GBM12 (proneural), GBM6 (classical), and U251 (Mesenchymal) Figure 6 (in vivo): GBM 43 (proneural)

Plating densities varied based on the specific well plates and dishes utilized. Timing of cultures for each experiment are listed in main text. Please see specifics on Seahorse experiments in Section 4.5.

### 4.2. Animals

Athymic nude mice (nu/nu; Charles River; Skokie, IL, USA) were housed according to all Institutional Animal Care and Use Committee (IACUC) guidelines and in compliance with all applicable federal and state statutes governing the use of animals for biomedical research. The specific protocol approval was IS00004080 (2 September 2016 to 21 September 2022). Briefly, animals were housed in shoebox cages with no more than 5 mice per cage in a temperature and humidity-controlled room. Food and water were available ad libitum. A strict 12 h light-dark cycle was maintained.

Intracranial implantation of glioblastoma cells was performed as previously published [58]. Briefly, animals received prophylactic injection of buprenex and Metacam via intraperitoneal (i.p.) injection, followed by an i.p injection of ketamine/xylazine anesthesia mixture (Henry Schien; New York, NY, USA). Sedation was confirmed by foot pinch. Artificial tears were applied to each eye and the scalp was sterilized repeatedly with betadine and ethanol. The scalp was then bisected using a scalpel to expose the skull. A drill was used to make a small burr hole above the right frontal lobe (approximately 1 mm in diameter). Animals were then placed into the stereotactic rig and a Hamilton syringe loaded with the cells was brought into the burr hole. The needle point was lowered 3 mm from the dura and injection of 5 μL of cell mixture took place over one minute. The needle was then raised slightly and left undisturbed for 1 min to ensure proper release of the cell mixture. After this minute, the syringe was carefully removed. The animal’s head position was maintained, and the skin of the scalp was closed with sutures (Ethicon; Cincinnati, OH, USA). Animals were then placed in fresh cages with circulating heat underneath and monitored for recovery. All instruments were sterilized with a bead sterilizer between animals and all other necessary procedures to maintain a sterile field were performed.

Drug treatments were initiated seven days after intracranial implantation. Animals received i.p. injections of either TMZ (2.5 mg/kg) or equimolar DMSO. Injections were performed daily for five consecutive days.

Animals were monitored daily for signs of sickness, including reduction in body weight, lowered body temperature, lack of grooming, hunched appearance, and behavioral signs by a blinded experimenter. Animals were euthanized when, in the opinion of the blinded experimenter, they would not survive until the next day. Animal sacrifices were performed according to Northwestern University guidelines. Briefly, animals were placed into CO_2_ chambers and the flow of CO_2_ was initiated; the flow rate did not exceed 2 L CO_2_/min while the animals were conscious. Whole brains were removed and washed in ice-cold phosphate buffer saline (PBS; Corning; Corning, NY, USA). For those brains utilized for FACS analysis, please see Flow Cytometry section of the Section 4.4.

### 4.3. RNA Isolation and MicroArray

After treatment, cells were dissociated with trypsin and washed with PBS. RNA extraction was performed using Qiagen’s RNeasy kit (Qiagen Inc.; Germantown, MD, USA) according to the manufacturer’s instructions. Quantification of RNA concentrations was performed using a NanoDrop 2000 (ThermoFisher; San Diego, CA, USA) and cDNA was synthesized according to established protocols using BioRad’s iScript kit using 1000 ng of total RNA per sample (BioRad; Hercules, CA, USA). The following cycles was used in a C1000 Thermal Cycler (BioRad) to synthesize cDNA: 5 min at 25 °C, 30 min at 42 °C, 5 min at 85 °C, and then temperature stabilized to 4 °C. RNA microanalysis was then performed using the Affymetrix 1300 platform.

### 4.4. Flow Cytometry Analysis

For in vitro experiments, cells were collected at serial time points after the beginning of treatment (days 2, 4, 6, 8), and fresh surface staining was performed. Live cells were collected and rinsed in PBS. For uptake experiments, cells were then incubated with APC anti-CD133 antibody and 2-NBDG. After 30 min incubation at 4 °C, cells were washed with PBS thoroughly. Pre-incubated QDOTs were then added to each well and allowed to incubate at room temperature for 4 min. After a final PBS wash, cells were analyzed using the flow cytometer.

In vivo studies began with the sacrifice of tumor bearing mice and immediate removal of the whole brain. Brains were washed in ice-cold PBS, and then bisected down the longitudinal fissure and right brains (tumor-bearing) were passed through a 70 µM strainer. These single cell suspensions were then incubated in ACK lysis buffer (Lonza; Walkersville, MA, USA) for 5 min at 20–25 °C to lysis any blood cells. After washing with PBS, cells were stained as in in vitro experiments. Human leukocyte antigen (HLA) staining was used to identify human tumor cells. All cells were collected in PBS supplemented with 1% bovine serum albumin (BSA; Fisher Scientific; Fair Lawn, NJ, USA) and sodium azide, and kept on ice until read.

The following antibodies were used: anti-HLA-PB (1:200; BioLegend, San Diego, CA, USA), anti-CD133-APC (2:100; Miltenyi Biotec, Auburn, CA, USA), 2-NDBG (Fisher), and QDOT (in house synthesis).

### 4.5. Seahorse Extracellular Fluid Analysis

Seahorse analysis was performed according to manufacturer’s guidance and in line with other published protocols [29,59]. GBM 39, 12 and 5 were lifted and adhered at a concentration of 0.5–1.5 × 10^5^ cells per well to XFe96 well culture plates (Agilent; Santa Clara, CA, USA) using CellTak tissue adhesive (Corning; Corning, NY, USA). We then performed a Mito-Stress test as described in the manufacturer’s protocol utilizing a Seahorse XFe96 well extracellular flux analyzer. Briefly, cells were attached and suspended in XF Base medium supplemented with 2 mM L-glutamine (Fisher; Hampton, NH, USA). After baseline ECAR reading, glucose was injected to a final concentration of 10 mM to determine glucose stimulated ECAR rate. Next, oligomycin (1 µM final concentration) was injected to determine maximal glycolytic capacity. Lastly, a competitive inhibitor of glycolysis (2-DG; 2-deoxyglucose) was injected (50 mM) to validate ECAR as a measure of glycolytic flux. To probe the role of fatty acid metabolism, Etomoxir (10 µM) and Palmatate (150 µM) were included. Critically, it was recently shown that Etomoxir used at concentrations above 100 µM has a range of off target effects [60], which we were well below. Data was analyzed using Agilent’s proprietary Wave software, Version 2.2.0.

### 4.6. Statistical Analysis

All statistical analyses were performed using the GraphPad Prism Software v4.0 (GraphPad Software, San Diego, CA, USA). Where applicable, one-way ANOVA, unpaired *t*-test, and log-rank test were applied. Survival distributions were estimated with the Kaplan–Meier method. A *p* value < 0.05 was considered statistically significant.

## 5. Conclusions

The ability to adapt to dynamic surroundings remains one of the key factors enabling GBM’s aggressive therapy resistance. These data suggest a novel mechanism by which GBM cells respond to microenvironmental cues; further, they highlight an emerging connection between cellular plasticity, the ability of GBM cells to attain multiple genetic and protein expression phenotypes, as well as metabolic behavior. Specifically, our results demonstrate that: (1) forcing GBM cells to adopt a specific cellular fate via alterations in environmental conditions induces concurrent changes in metabolic behavior; (2) chemotherapeutic stress, already established as capable of influencing cellular plasticity, causes GBM cells to increase their reliance on lipid metabolism; (3) alkylation chemotherapy increases the expression of several genes known to regulate and drive lipid metabolism; and (4) therapeutic stress causes GBM cells, especially GSCs, to aggressively uptake fatty acids from their environment, both in vitro and in vivo. These results provide evidence linking microenvironmental cues, such as therapeutic stress, and altering metabolic function in order to promote cellular adaptation. These novel connections may provide new avenues for enhancing the effectiveness of current treatment modalities and developing novel, targeted therapies for GBM. Put more simply, this strategy might rob GBM cells of the raw material required for adaptation. Future research will shed more light on this avenue and, hopefully, provide desperately needed therapeutics for patients battling GBM.

## Figures and Tables

**Figure 1 cancers-12-03126-f001:**
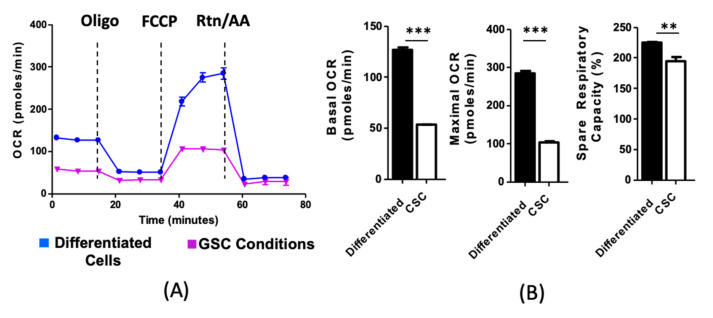
Glioma stem cells (GSCs) exhibit unique metabolic phenotypes, including general quiescence and increased fatty acid uptake. (**A**) Representative tracing of metabolic analysis. Patient derived xenograft (PDX) glioblastoma (GBM) 43 cells were cultured as either adherent monocultures in forced differentiation media (Dulbecco’s Modified Eagle Medium (DMEM) supplemented with 1% Fetal Bovine Serum (FBS)) or GSC-promoting media (Serum-free Neurobasal media with Epidermal Growth Factor (EGF) and Fibroblast Growth Factor (FGF)). After 72 h, cells were subjected to Seahorse metabolic analysis. (**B**) Analysis reveals that cells cultured in stemness-promoting conditions exhibit reduced basal and maximal oxygen consumption rates (OCR), as well as spare respiratory capacity. Bars represent means of three experiments. Error bars show standard deviation. Comparison was performed using Student’s *t*-test. ** *p* < 0.01 *** *p* < 0.001.

**Figure 2 cancers-12-03126-f002:**
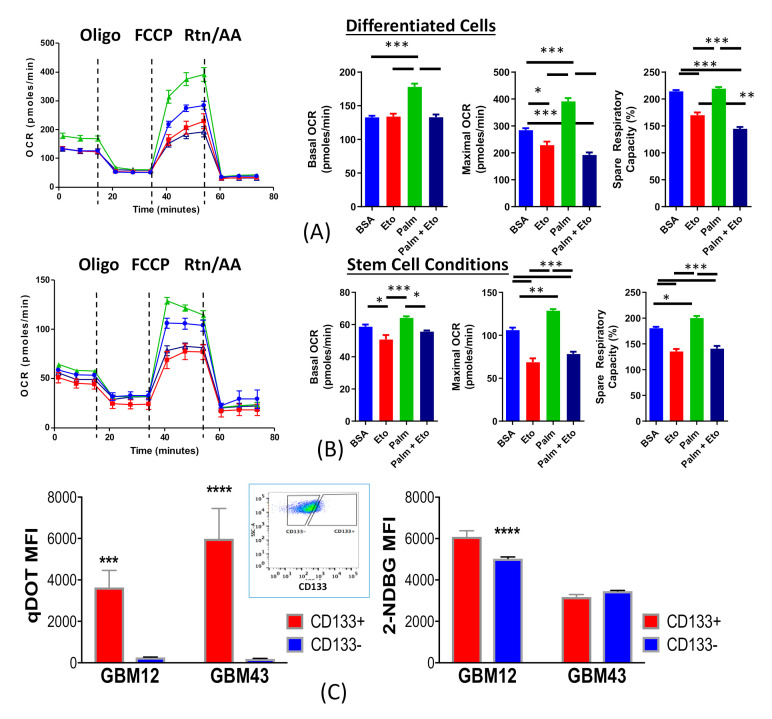
Glioma cells cultured in GSC conditions utilize both Endogenous and Exogenous FA Metabolism. (**A**) Patient derived GBM43 cells were cultured in differentiation promoting media and metabolic phenotype was analyzed by seahorse after 72 h. In order to assess the reliance on fatty acids, cells were plated in the presence of etomoxir (Eto), a mitochondrial fatty acid uptake inhibitor, 150 µm palmitate (Palm), a major exogenous fatty acid, bovine serum albumin, a control, or a combination of Palm and Eto. Analysis indicates that differentiated cells rely largely on exogenous sources of fatty acid. (**B**) PDX GBM43 cells grown in stemness promoting media were analyzed in the same manner, indicating that they utilize both endogenous and exogenous sources of fatty acid. (**C**) In order to assess the difference in metabolism in GSCs and non-GSCs growing in a mixed culture, we utilized 2-NDBG, a fluorescent analogue of glucose, and our proprietary quantum dots conjugated to palmitic acid. FACS analysis of PDX GBM43 and PDX GBM12 cells growing in neutral media (DMEM with 1%FBS) showed that GSCs, identified by assessment of CD133 expression (inset graph), uptake a significantly higher level of fatty acids relative to their non-GSC counterparts (as measured by MFI within each population gate). A slight difference in glucose uptake was observed in GBM12 cells. Bars represent means of three experiments. Error bars show standard deviation. One-way ANOVA with Tukey’s Post-Hoc with multiple comparisons. * *p* < 0.05, ** *p* < 0.01, *** *p* < 0.001, **** *p* < 0.0001.

**Figure 3 cancers-12-03126-f003:**
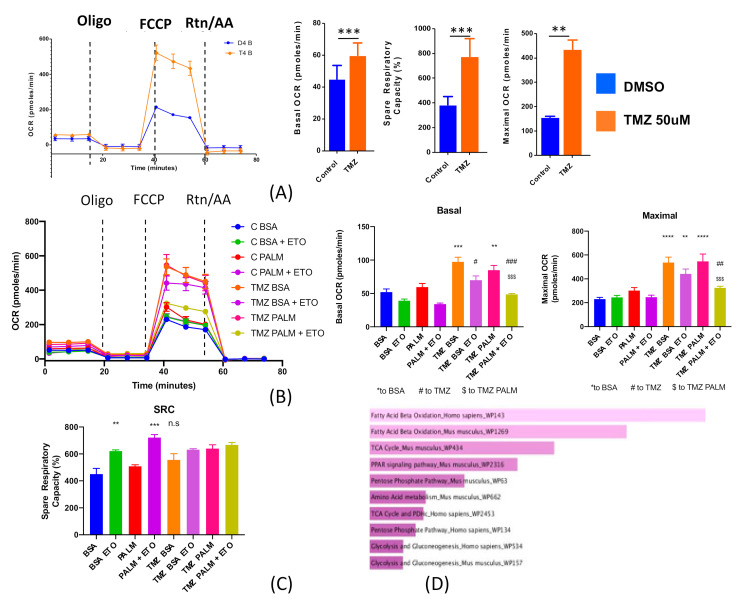
TMZ increases mitochondrial metabolism and lipid uptake. (**A**) PDX GBM43 cells were treated with temozolomide (TMZ, 50 μΜ) or equimolar DMSO for 96 h and then subjected to extracellular flux analysis. TMZ treated cells exhibited significantly heightened metabolism, as assessed by basal and maximal oxygen consumption rate (OCR) and spare respiratory capacity (SRC). (**B/C**) In order to understand the contribution of fatty acids to this phenotype, we repeated the experiment in the presence of 150 µm palmitate as well as Etomoxir. Addition of this exogenous fatty acid did not further heighten metabolism, while etomoxir reduced TMZ-driven increases in metabolism. (**D**) Microarray analysis of TMZ-treated PDX GBM43 cells reveals that chemotherapy increases the expression of genes related to fatty acid oxidation and the citric acid cycle. Bars show mean of three independent experiments, and error bars show SD. In (**A**) comparison was performed using Student’s *t*-test. In (**B**) one-way ANOVA was utilized with multiple comparisons. ^n.s^
*p* > 0.05, ** *p* < 0.01, *** *p* < 0.001, **** *p* < 0.0001, ^##^
*p* < 0.01, ^###^
*p* < 0.001 relative to TMZ; ^$$$^
*p* < 0.001 to TMZ + PALM.

**Figure 4 cancers-12-03126-f004:**
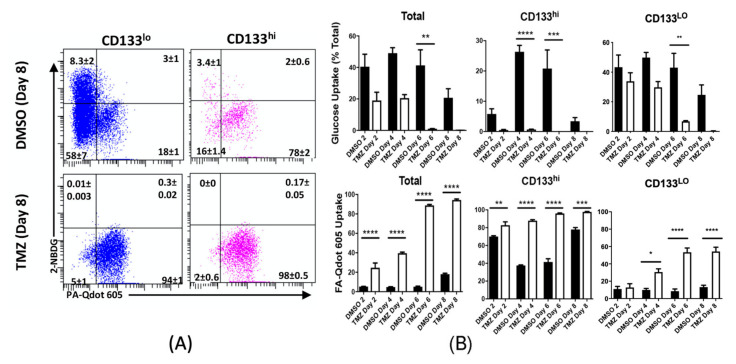
Chemotherapy increases uptake of fatty acids in GBM cells. Patient derived xenograft GBM43 cells were treated with TMZ (50 μM) or equimolar DMSO for 2, 4, 6, or 8 days. Uptake of fatty acids and glucose were assessed in live cells using our qDOT-palmitate system and 2-NDBG. GSCs were identified by the cell surface marker CD133. (**A**) Representative FACS plots for fatty acid and glucose uptake from each treatment group eight days following treatment. Cells were divided into CD133 high and low based on antibody staining and unstained controls. (**B**) Mean fatty acid uptake and glucose, based on percentage of cells positive for each fluorescent analogue, were calculated from three separate experiments. All live cells, regardless of CD133 status (total, representative image in Appendix A) were analyzed, as were GSCs specifically (CD133hi). TMZ increased fatty acid uptake in the total cell population, while GSCs tended to have high fatty acid uptake, regardless of treatment status. Bars represent mean of three independent experiments. Error bars indicate standard error measure (SEM). Values were compared with one-way ANOVA with multiple comparisons. * *p* < 0.05, ** *p* < 0.01, *** *p* < 0.001, and **** *p* < 0.0001.

**Figure 5 cancers-12-03126-f005:**
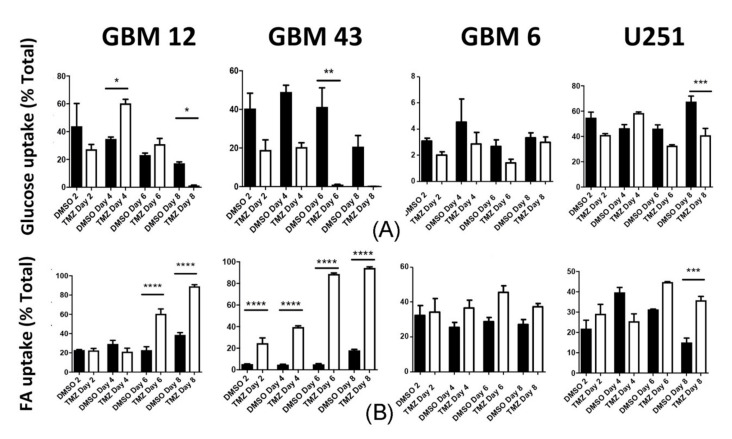
Therapeutic stress induced a metabolic shift associated with lipid uptake in multiple glioma cell lines. Multiple GBM cell lines were treated with TMZ (50 μM) or equimolar DMSO for 2, 4, 6, or 8 days, after which uptake of (**A**) glucose and (**B**) fatty acid were analyzed via FACS analysis, as in Figure 4. Appendix A provide further detail on these experiments. Bars represent mean of three independent experiments. Error bars indicate standard error measure (SEM). Values were compared with one-way ANOVA with multiple comparisons followed by Tukey’s post-hoc test for individual comparisons. * *p* < 0.05, ** *p* < 0.01, *** *p* < 0.001, and **** *p* < 0.0001.

**Figure 6 cancers-12-03126-f006:**
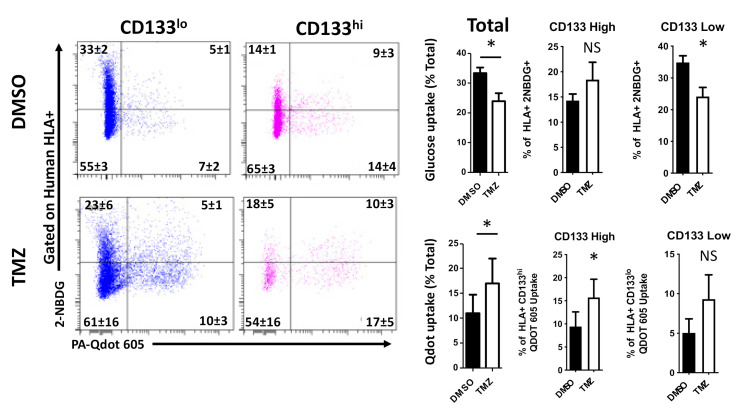
Therapeutic stress induces metabolic changes in GBM tumors in vivo. Athymic nude mice had 150,000 GBM43 cells implanted intracranially. Following one week of engraftment, mice were treated daily with either DMSO or TMZ (2.5 mg/kg) for five consecutive days. Once mice began to show symptoms of tumor burden (weight loss, hemiparesis), they were sacrificed and live cells from the right hemisphere of the brain was collected and analyzed for uptake of fatty acid and glucose as before. FACS analysis for human cells (Human leukocyte antigen+) and GSCs (CD133+) was performed simultaneously. Treated me TMZ led to an increase in fatty acid uptake. Gating on CD133 high cells revealed a subtype specific increase in fatty acid uptake (representative image for all cells in Appendix A). Bars represent means from three experiments. Error bars show the standard deviation. Comparisons were done using a student *t*-test. * *p* < 0.05.

**Table 1 cancers-12-03126-t001:** Key genetic attributes of patient derived xenographs (PDX) cell lines.

Method	Mayo Sarkaria qMS-PCR	MD Anderson CLIA qMS-PCR	Sanger Sequencing		
GBM	MGMT METHYLATION	MGMT METHYLATION	TERT	IDH1	IDH2
6	U	Indeterminate	C228T	wt	wt
12	M	M	C250T	wt	wt
43	U	M	C228T	wt	wt

Cell lines used in experiments are all IDH1/2 wildtype. They have two distinct TERT mutations. U = unmethylated, M = methylated.

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
