# Peer review of "Temozolomide Treatment Increases Fatty Acid Uptake in Glioblastoma Stem Cells"

_cancers, 2020, doi:10.3390/cancers12113126_

Round 1

Reviewer 1 Report

The authors have appropriately answered almost all of the reviewer’s questions and have improved the manuscript significantly.

Several issues should be considered before publication.

  1. Please proofread the text for any mistakes/misprints. Line 370: “U251 human glioma cell lines…”. Have you used several types of U251 cells? If yes, please add the description. If not, the please correct the grammar.
  2. Materials and Methods. Lines 378-379: If that image is the Table, it needs an appropriate legend. If it is taken from some other source, please add the reference.
  3. Line 437: “NanoDrop (ThermoFisher)” – please add proper equipment description (model, company name, city, country).
  4. Please check the Supplementary materials, the sentence there shows the previous title: “Supplementary materials for “Chemotherapeutic stress alters metabolic behavior in glioblastoma stem cells””.

As glioblastoma remains uncurable regardless of any modern therapies, any experiments that can support further therapy development deserve publishing.

Reviewer 2 Report

In the manuscript entitled “Temozolomide treatment increases fatty acid uptake in glioblastoma stem cells” the authors responded to the several issues raised by the reviewer in the first version of the manuscript. There are only a few points that the authors should address. - The abstract: the authors should modify the conclusion sentence based on the obtained results (for example, temozolomide-induced instead of therapy-induced) - The representative image in fig. 4 and Fig.6 did not match with the values in the graphic, please check the correspondence. Furthermore, the authors state that quantify the staining in total (regardless of CD133) and CD133 high. The author must report the representative image of the experiments of total cells. - The authors must report the quantitative analysis of CD133lo (as reported in Fig. 6) in comparison to the CD133high and Total

Reviewer 3 Report

Dear Authors,

I am glad to see your answers. Need some changes in the language.

Regards,

Pranjal Sarma

Round 2

Reviewer 2 Report

The manuscript is now suitable for pubblication

This manuscript is a resubmission of an earlier submission. The following is a list of the peer review reports and author responses from that submission.

Round 1

Reviewer 1 Report

Reviewer’s comments

The article by Caragher et al. describes the experiments showing therapy-induced changes in glioblastoma stem cells that can play a role in new therapy strategies. A series of extensive experiments have been carried out by the authors of the manuscript to support their ideas. As malignant gliomas and particularly glioblastoma multiforme (GBM) remain uncurable regardless of any modern therapies, and radiation and temozolomide can slightly prolong patient overall survival, any experiments that can support further therapy development deserve publishing. However, it is difficult to accept the manuscript in the present form, and the following issues are recommended to be clarified by the authors before further evaluation:

  1. The title represents quite a general and already known concept that has been covered in a number of recent research articles and reviews [1-6]. More specified title related to the carried experiments (fatty acid intake, TMZ, etc.) might be more advantageous to show the novelty of the study, otherwise justified.
  2. The novelty of the study should be more clearly explained in relation to already published recent reports in the corresponding field [1, 4-6].
  3. Analyzing more recent reports on the topic of the research might be beneficial to enrich the Introduction and Discussion sections, as a number of recent studies bring valuable concepts that might influence the interpretation of the results in the current study [1-11].
  4. The main hypothesis needs to be clearly formulated excluding already known general concepts (lines 55-59).
  5. The Results section and Figure legends contain a lot of method descriptions.
  6. Though it might be clear from the figures, it is better to clearly indicate in the Methods which patient-derived cells were used for which experiment. Typically, the ethics committee is required for animal experiments or any equivalent, otherwise justified the absence of such a committee.
  7. The essence of Seahorse analysis should be explained to the readers unfamiliar with the method.
  8. Referring the patient-obtained glioma cells into the molecular subtypes (proneural, mesenchymal, classical) might be better explained and/or referenced.
  9. It might be generally advantageous for the article to have the information orderly placed in the corresponding sections of the article.
  10. The Conclusion section reminds Discussion and should be better rewritten to provide concise conclusive statements to support the hypothesis. Paragraph 1 in the Discussion (lines 283-292) would fit better in the conclusion. Limitations of the study should be included in the discussion, if any considered.
  11. The way of describing the origins of the reagents, kits, software, etc. should be unified.
  12. The article should be proofread before further evaluation (“One key questions”: line 310; references on lines 368, 385, “heighted” metabolism: line 197, the subscripts and superscripts should be corrected (CO2, 105, etc.): lines 371, 407-408, 449, etc.).

References:

  1. Zhou W, Wahl DR. Metabolic Abnormalities in Glioblastoma and Metabolic Strategies to Overcome Treatment Resistance. Cancers (Basel). 2019 Aug 23;11(9):1231. doi: 10.3390/cancers11091231. PMID: 31450721; PMCID: PMC6770393.
  2. Kant, S., Kesarwani, P., Prabhu, A. et al. Enhanced fatty acid oxidation provides glioblastoma cells metabolic plasticity to accommodate to its dynamic nutrient microenvironment. Cell Death Dis 11, 253 (2020). https://doi.org/10.1038/s41419-020-2449-5
  3. Garnier D, Renoult O, Alves-Guerra MC, Paris F, Pecqueur C. Glioblastoma Stem-Like Cells, Metabolic Strategy to Kill a Challenging Target. Front Oncol. 2019;9:118. Published 2019 Mar 6. doi:10.3389/fonc.2019.00118
  4. Badr CE, Silver DJ, Siebzehnrubl FA, Deleyrolle LP. Metabolic heterogeneity and adaptability in brain tumors [published online ahead of print, 2020 Jun 6]. Cell Mol Life Sci. 2020;10.1007/s00018-020-03569-w. doi:10.1007/s00018-020-03569-w
  5. Masui K, Onizuka H, Cavenee WK, Mischel PS, Shibata N. Metabolic reprogramming in the pathogenesis of glioma: Update. Neuropathology. 2019;39(1):3-13. doi:10.1111/neup.12535
  6. Tiek, D.M., Rone, J.D., Graham, G.T. et al. Alterations in Cell Motility, Proliferation, and Metabolism in Novel Models of Acquired Temozolomide Resistant Glioblastoma. Sci Rep 8, 7222 (2018). https://doi.org/10.1038/s41598-018-25588-1
  7. Libby CJ, Tran AN, Scott SE, Griguer C, Hjelmeland AB. The pro-tumorigenic effects of metabolic alterations in glioblastoma including brain tumor initiating cells. Biochim Biophys Acta Rev Cancer. 2018 Apr;1869(2):175-188. doi: 10.1016/j.bbcan.2018.01.004. Epub 2018 Jan 31. PMID: 29378228; PMCID: PMC6596418.
  8. Ebert, D.; Haller, R.G.; Walton, M.E. Energy contribution of octanoate to intact rat brain metabolism measured by 13C nuclear magnetic resonance spectroscopy. J. Neurosci.2003,23, 5928–5935. https://doi.org/10.1523/JNEUROSCI.23-13-05928.2003
  9. Kucharzewska P, Christianson HC, Belting M (2015) Global Profiling of Metabolic Adaptation to Hypoxic Stress in Human Glioblastoma Cells. PLoS ONE 10(1): e0116740. https://doi.org/10.1371/journal.pone.0116740
  10. Zhang K, Xu P, Sowers JL, et al. Proteome Analysis of Hypoxic Glioblastoma Cells Reveals Sequential Metabolic Adaptation of One-Carbon Metabolic Pathways. Mol Cell Proteomics. 2017;16(11):1906-1921. doi:10.1074/mcp.RA117.000154
  11. Cheng X, Geng F, Pan M, et al. Targeting DGAT1 Ameliorates Glioblastoma by Increasing Fat Catabolism and Oxidative Stress. Cell Metab. 2020;32(2):229-242.e8. doi:10.1016/j.cmet.2020.06.002

Author Response

Reviewer 1

The article by Caragher et al. describes the experiments showing therapy-induced changes in glioblastoma stem cells that can play a role in new therapy strategies. A series of extensive experiments have been carried out by the authors of the manuscript to support their ideas. As malignant gliomas and particularly glioblastoma multiforme (GBM) remain uncurable regardless of any modern therapies, and radiation and temozolomide can slightly prolong patient overall survival, any experiments that can support further therapy development deserve publishing. However, it is difficult to accept the manuscript in the present form, and the following issues are recommended to be clarified by the authors before further evaluation:

Thank you very much for all of your comments and suggestion. We are particularly grateful for the references you recommended. We have included them all and hope that their inclusion has sharpened the presentation of our data and expanded their potential importance to the field of GBM metabolic plasticity. Please see point-by-point discussion of each of your points below.

1. The title represents quite a general and already known concept that has been covered in a number of recent research articles and reviews [1-6]. More specified title related to the carried experiments (fatty acid intake, TMZ, etc.) might be more advantageous to show the novelty of the study, otherwise justified.

Response: Thank you for this suggestion. We apologize that the previous title was overly broad. It has been adjusted to “Temozolomide treatment increases fatty acid uptake in glioblastoma stem cells”. We hope that this new title more accurately describes the key take-away of our paper.

We have also incorporated the references that you recommended in the paper. We are grateful for these references. 

2. The novelty of the study should be more clearly explained in relation to already published recent reports in the corresponding field [1, 4-6].

Response: Thank you again for these references. They have been incorporated. For example (Line 62-70):

To that end, a range of phenotypic domains have been examined for contributions to tumor plasticity during therapy, including epigenetic modification and gene expression [1,2]. Recently, the metabolic phenotype of GBM tumors has emerged as an exciting area of research (expertly reviewed [3,4]). GBM tumors have been shown to rely on glycolysis during phases of growth [5], with inhibition of glycolysis improving survival of mouse models of GBM[6,7]. Other groups have highlighted the importance of fatty acid oxidation in GBM growth[8]. Critically, metabolism in brain tumors is heterogeneous and linked to stemness. For example, a GSC population has been demonstrated to exhibit elevated lipid levels and utilize fatty acids to withstand nutrient deprivation[9]. Other reports have highlighted the importance of fatty acid synthesis in enabling adoption of a GSC state[10].

3. Analyzing more recent reports on the topic of the research might be beneficial to enrich the Introduction and Discussion sections, as a number of recent studies bring valuable concepts that might influence the interpretation of the results in the current study [1-11].

These references have been incorporated. Thank you for them. Please see point 2 for specific text changes.

4. The main hypothesis needs to be clearly formulated excluding already known general concepts (lines 55-59).

Response: The hypothesis has been reformulated, in conjunction with re-working the introduction more broadly. It now reads (Lines 75-80):

“These exciting reports raise the question of how chemotherapeutic stress specifically alters GBM metabolic behavior, plasticity, and therapy resistance. We sought to probe this question and hypothesized that standard of care TMZ treatment would alter the metabolic behavior of GBM cells in vitro and in vivo.”

5. The Results section and Figure legends contain a lot of method descriptions.

Response: We have reduced the description of the methods throughout the manuscript. Now methods are only described in detail when they are first mentioned. Figure legends have also been shortened.

6. Though it might be clear from the figures, it is better to clearly indicate in the Methods which patient-derived cells were used for which experiment. Typically, the ethics committee is required for animal experiments or any equivalent, otherwise justified the absence of such a committee.

Response: The specific cell lines for each experiment have been included in the methods (Lines 631-632):

Figure 1-4: GBM43 (proneural)

            Figure 5: GBM43, GBM12 (proneural), GBM6 (classical), and U251 (Mesenchymal)

            Figure 6 (in vivo): GBM 43 (proneural)

We have added more information on our IACUC approval and dates (please see section 4.2 on animal use). Text reads:

Response: The specific protocol approval was IS00004080 (9/2/2016 to 9/21/2022).

7. The essence of Seahorse analysis should be explained to the readers unfamiliar with the method.

Response: Thank you for this suggestion. We have added a brief description and citation for Seahorse in the results section when it is first encountered, a longer background description in the methods section, as well as some references for interested readers[11,12].

8. Referring the patient-obtained glioma cells into the molecular subtypes (proneural, mesenchymal, classical) might be better explained and/or referenced.

Response: We have added more details on subtypes in section 2.4 of the results. We have also added a longer description and references in the methods section, including key citations [13,14]. (Lines 411-413):

Of note, GBM tumors are broken into several subtypes based on their genetic expression profiles. These subtypes include classical, proneural, and mesenchymal[14]. These subtypes are associated with different driver mutations, responses to therapy, and prognosis[15].

9. It might be generally advantageous for the article to have the information orderly placed in the corresponding sections of the article.

Response: Thank you for this comment. As mentioned above, descriptions of methods have been increased in the methods section. In the results, methods are described once in moderate detail when first mentioned and then referred to in brief when further mentioned.

10. The Conclusion section reminds Discussion and should be better rewritten to provide concise conclusive statements to support the hypothesis. Paragraph 1 in the Discussion (lines 283-292) would fit better in the conclusion. Limitations of the study should be included in the discussion, if any considered.

Response: Thank you for this input. We have reworked the discussion and conclusion with your suggestions. Limitations of the study are described and analyzed as their own paragraph in the discussion (Lines 589-602):

This study is not without limitations. First, the technical difficulty of culturing large number of GSCs limited our ability to probe a confirmed post-TMZ sample of GSCs by Seahorse. Second, GBM tumor cells are exquisitely sensitive to changes in culture conditions[16,17]. While we were able to confirm our major finding in a mouse model, there is always concern that findings may not perfectly recapitulate the physiological conditions of GBM tumors in human patients. Third, we only investigated a specific subtype of fatty acids. Lipid metabolism in cancer is an expansive area of research [18-20]; we hope that further exploration will reveal further details about the connection between specific lipid particles and GBM plasticity. Finally, these experiments were conducted in cell lines that are all isocitrate dehydrogenase (IDH1/2) wild type (see methods). IDH status is a key driver of differential prognosis in GBM [21,22]. As such, these data can only be applied to IDH wildtype cell lines and tumors. Further research will establish the role of IDH mutations in GBM therapy-induced plasticity. Despite these limitations, this study highlights the importance of metabolic plasticity in GBM and its potential as a therapeutic target for increasing the efficacy of TMZ.

11. The way of describing the origins of the reagents, kits, software, etc. should be unified.

Response: We appreciate the suggestion. We have unified our system for method descriptions.

12. The article should be proofread before further evaluation (“One key questions”: line 310; references on lines 368, 385, “heighted” metabolism: line 197, the subscripts and superscripts should be corrected (CO2, 105, etc.): lines 371, 407-408, 449, etc.).

 Response: Thank you for pointing out these errors. We regret their inclusion in the first draft. They  have each been addressed.

We have attached the revised manuscript with yellow highlight in the text to indicating all the changes.

REFERENCES

  1. Liau, B.B.; Sievers, C.; Donohue, L.K.; Gillespie, S.M.; Flavahan, W.A.; Miller, T.E.; Venteicher, A.S.; Hebert, C.H.; Carey, C.D.; Rodig, S.J., et al. Adaptive Chromatin Remodeling Drives Glioblastoma Stem Cell Plasticity and Drug Tolerance. Cell Stem Cell 2017, 20, 233-246 e237, doi:10.1016/j.stem.2016.11.003.
  2. Heddleston, J.M.; Li, Z.; McLendon, R.E.; Hjelmeland, A.B.; Rich, J.N. The hypoxic microenvironment maintains glioblastoma stem cells and promotes reprogramming towards a cancer stem cell phenotype. Cell Cycle 2009, 8, 3274-3284, doi:10.4161/cc.8.20.9701.
  3. Badr, C.E.; Silver, D.J.; Siebzehnrubl, F.A.; Deleyrolle, L.P. Metabolic heterogeneity and adaptability in brain tumors. Cell Mol Life Sci 2020, 10.1007/s00018-020-03569-w, doi:10.1007/s00018-020-03569-w.
  4. Masui, K.; Onizuka, H.; Cavenee, W.K.; Mischel, P.S.; Shibata, N. Metabolic reprogramming in the pathogenesis of glioma: Update. Neuropathology 2019, 39, 3-13, doi:10.1111/neup.12535.
  5. Marin-Valencia, I.; Yang, C.; Mashimo, T.; Cho, S.; Baek, H.; Yang, X.L.; Rajagopalan, K.N.; Maddie, M.; Vemireddy, V.; Zhao, Z., et al. Analysis of tumor metabolism reveals mitochondrial glucose oxidation in genetically diverse human glioblastomas in the mouse brain in vivo. Cell Metab 2012, 15, 827-837, doi:10.1016/j.cmet.2012.05.001.
  6. Tateishi, K.; Iafrate, A.J.; Ho, Q.; Curry, W.T.; Batchelor, T.T.; Flaherty, K.T.; Onozato, M.L.; Lelic, N.; Sundaram, S.; Cahill, D.P., et al. Myc-Driven Glycolysis Is a Therapeutic Target in Glioblastoma. Clin Cancer Res 2016, 22, 4452-4465, doi:10.1158/1078-0432.CCR-15-2274.
  7. Sanzey, M.; Abdul Rahim, S.A.; Oudin, A.; Dirkse, A.; Kaoma, T.; Vallar, L.; Herold-Mende, C.; Bjerkvig, R.; Golebiewska, A.; Niclou, S.P. Comprehensive analysis of glycolytic enzymes astherapeutic targets in the treatment of glioblastoma. PLoS One 2015, 10, e0123544, doi:10.1371/journal.pone.0123544.
  8. Lin, H.; Patel, S.; Affleck, V.S.; Wilson, I.; Turnbull, D.M.; Joshi, A.R.; Maxwell, R.; Stoll, E.A. Fatty acid oxidation is required for the respiration and proliferation of malignant glioma cells. Neuro Oncol 2017, 19, 43-54, doi:10.1093/neuonc/now128.
  9. Hoang-Minh, L.B.; Siebzehnrubl, F.A.; Yang, C.; Suzuki-Hatano, S.; Dajac, K.; Loche, T.; Andrews, N.; Schmoll Massari, M.; Patel, J.; Amin, K., et al. Infiltrative and drug-resistant slow-cycling cells support metabolic heterogeneity in glioblastoma. EMBO J 2018, 37, doi:10.15252/embj.201798772.
  10. Yasumoto, Y.; Miyazaki, H.; Vaidyan, L.K.; Kagawa, Y.; Ebrahimi, M.; Yamamoto, Y.; Ogata, M.; Katsuyama, Y.; Sadahiro, H.; Suzuki, M., et al. Inhibition of Fatty Acid Synthase Decreases Expression of Stemness Markers in Glioma Stem Cells. PLoS One 2016, 11, e0147717, doi:10.1371/journal.pone.0147717.
  11. Zhang, J.; Zhang, Q. Using Seahorse Machine to Measure OCR and ECAR in Cancer Cells. Methods Mol Biol 2019, 1928, 353-363, doi:10.1007/978-1-4939-9027-6_18.
  12. Yepez, V.A.; Kremer, L.S.; Iuso, A.; Gusic, M.; Kopajtich, R.; Konarikova, E.; Nadel, A.; Wachutka, L.; Prokisch, H.; Gagneur, J. OCR-Stats: Robust estimation and statistical testing of mitochondrial respiration activities using Seahorse XF Analyzer. PLoS One 2018, 13, e0199938, doi:10.1371/journal.pone.0199938.
  13. Giannini, C.; Sarkaria, J.N.; Saito, A.; Uhm, J.H.; Galanis, E.; Carlson, B.L.; Schroeder, M.A.; James, C.D. Patient tumor EGFR and PDGFRA gene amplifications retained in an invasive intracranial xenograft model of glioblastoma multiforme. Neuro-oncology 2005, 7, 164-176.
  14. Verhaak, R.G.; Hoadley, K.A.; Purdom, E.; Wang, V.; Qi, Y.; Wilkerson, M.D.; Miller, C.R.; Ding, L.; Golub, T.; Mesirov, J.P., et al. Integrated genomic analysis identifies clinically relevant subtypes of glioblastoma characterized by abnormalities in PDGFRA, IDH1, EGFR, and NF1. Cancer Cell 2010, 17, 98-110, doi:10.1016/j.ccr.2009.12.020.
  15. Thakkar, J.P.; Dolecek, T.A.; Horbinski, C.; Ostrom, Q.T.; Lightner, D.D.; Barnholtz-Sloan, J.S.; Villano, J.L. Epidemiologic and molecular prognostic review of glioblastoma. Cancer Epidemiol Biomarkers Prev 2014, 23, 1985-1996, doi:10.1158/1055-9965.EPI-14-0275.
  16. Strickland, M.; Stoll, E.A. Metabolic Reprogramming in Glioma. Front Cell Dev Biol 2017, 5, 43, doi:10.3389/fcell.2017.00043.
  17. Caragher, S.; Chalmers, A.J.; Gomez-Roman, N. Glioblastoma's Next Top Model: Novel Culture Systems for Brain Cancer Radiotherapy Research. Cancers (Basel) 2019, 11, doi:10.3390/cancers11010044.
  18. Carracedo, A.; Cantley, L.C.; Pandolfi, P.P. Cancer metabolism: fatty acid oxidation in the limelight. Nat Rev Cancer 2013, 13, 227-232, doi:10.1038/nrc3483.
  19. Rohrig, F.; Schulze, A. The multifaceted roles of fatty acid synthesis in cancer. Nat Rev Cancer 2016, 16, 732-749, doi:10.1038/nrc.2016.89.
  20. Griffin, J.L.; Shockcor, J.P. Metabolic profiles of cancer cells. Nat Rev Cancer 2004, 4, 551-561, doi:10.1038/nrc1390.
  21. Huse, J.T.; Holland, E.C. Targeting brain cancer: advances in the molecular pathology of malignant glioma and medulloblastoma. Nat Rev Cancer 2010, 10, 319-331, doi:10.1038/nrc2818.
  22. SongTao, Q.; Lei, Y.; Si, G.; YanQing, D.; HuiXia, H.; XueLin, Z.; LanXiao, W.; Fei, Y. IDH mutations predict longer survival and response to temozolomide in secondary glioblastoma. Cancer Sci 2012, 103, 269-273, doi:10.1111/j.1349-7006.2011.02134.x.
  23. Garnier, D.; Renoult, O.; Alves-Guerra, M.C.; Paris, F.; Pecqueur, C. Glioblastoma Stem-Like Cells, Metabolic Strategy to Kill a Challenging Target. Front Oncol 2019, 9, 118, doi:10.3389/fonc.2019.00118.
  24. Podergajs, N.; Brekka, N.; Radlwimmer, B.; Herold-Mende, C.; Talasila, K.M.; Tiemann, K.; Rajcevic, U.; Lah, T.T.; Bjerkvig, R.; Miletic, H. Expansive growth of two glioblastoma stem-like cell lines is mediated by bFGF and not by EGF. Radiol Oncol 2013, 47, 330-337, doi:10.2478/raon-2013-0063.
  25. Stockhausen, M.T.; Kristoffersen, K.; Stobbe, L.; Poulsen, H.S. Differentiation of glioblastoma multiforme stem-like cells leads to downregulation of EGFR and EGFRvIII and decreased tumorigenic and stem-like cell potential. Cancer Biol Ther 2014, 15, 216-224, doi:10.4161/cbt.26736.
  26. Muroski, M.E.; Miska, J.; Chang, A.L.; Zhang, P.; Rashidi, A.; Moore, H.; Lopez-Rosas, A.; Han, Y.; Lesniak, M.S. Fatty Acid Uptake in T Cell Subsets Using a Quantum Dot Fatty Acid Conjugate. Sci Rep 2017, 7, 5790, doi:10.1038/s41598-017-05556-x.
  27. Wang, D.; Green, M.F.; McDonnell, E.; Hirschey, M.D. Oxygen flux analysis to understand the biological function of sirtuins. Methods Mol Biol 2013, 1077, 241-258, doi:10.1007/978-1-62703-637-5_16.
  28. van der Windt, G.J.W.; Chang, C.H.; Pearce, E.L. Measuring Bioenergetics in T Cells Using a Seahorse Extracellular Flux Analyzer. Curr Protoc Immunol 2016, 113, 3 16B 11-13 16B 14, doi:10.1002/0471142735.im0316bs113.
  29. Nicholas, D.; Proctor, E.A.; Raval, F.M.; Ip, B.C.; Habib, C.; Ritou, E.; Grammatopoulos, T.N.; Steenkamp, D.; Dooms, H.; Apovian, C.M., et al. Advances in the quantification of mitochondrial function in primary human immune cells through extracellular flux analysis. PLoS One 2017, 12, e0170975, doi:10.1371/journal.pone.0170975.
  30. Nicholls, D.G.; Darley-Usmar, V.M.; Wu, M.; Jensen, P.B.; Rogers, G.W.; Ferrick, D.A. Bioenergetic profile experiment using C2C12 myoblast cells. J Vis Exp 2010, 10.3791/2511, doi:10.3791/2511.
  31. Huynh, F.K.; Green, M.F.; Koves, T.R.; Hirschey, M.D. Measurement of fatty acid oxidation rates in animal tissues and cell lines. Methods Enzymol 2014, 542, 391-405, doi:10.1016/B978-0-12-416618-9.00020-0.

Reviewer 2 Report

In the manuscript entitled “Chemotherapeutic stress alters metabolic behavior in glioblastoma stem cells”, the authors performed a characterization of lipid and glucose uptake in different PDX GBM cells and after the treatment with TMZ (the chemotherapeutic agent used in clinical practice). The authors reported the increase of fatty acid uptake based on TMZ treatment, the GBM cell phenotype, and the amount of stem cell counterpart identified as CD133 + cells. Despite the characterization of the metabolic shift in GBM cells after TMZ treatment could be of interest, the manuscript has several criticisms, and in the present form is not suitable for publication in Cancers journal.

The main criticisms:

  • The statistical analysis of the data is missing throughout the manuscript and in supplementary figures.
  • The introduction must be improved, several data on GBM and GSC metabolism has been reported (see the review Front Cell Dev Biol. 2017 Apr 26;5:43. doi: 10.3389/fcell.2017.00043. eCollection 2017; Neuropathology. 2019 Feb;39(1):3-13. doi: 10.1111/neup.12535; Front Oncol. 2019 Mar 6;9:118. doi: 10.3389/fonc.2019.00118. Article: Cell Death Dis 11, 253 (2020). https://doi.org/10.1038/s41419-020-2449-5).
  • IDH1 mutation is the main genetic mutation driving GBM development (see the above review). The isocitrate dehydrogenase enzymes are involved in the oxidative carboxylation of isocitrate to α‐ketoglutarate (α‐KG) in the cytosol and mitochondria. Of note, mutant IDH is relevant to glioma pathogenesis through its association with metabolic reprogramming. Based on the manuscript rationale, the authors must report the IDH1/2 status of all the used cell lines.
  • The methods used for the experiments are messy. The author should reorganize the section in several parts reporting for each experiment the time of in vitro culturing, the cell density used, and the media used. All this information is pivotal to interpreter the reported data.
  • Line 65: please specify the protocol used for CSC enrichment. Furthermore, similar data have been already reported in the literature (Proc Natl Acad Sci U S A. 2011 Sep 20; 108(38): 16062–16067. doi: 10.1073/pnas.1106704108)
  • The reference to Fig. S1 and Fig S2 are missing.
  • The authors used the CD133 as a marker to select the CSC subpopulation. In GBM many studies rely on cell surface markers such as CD133, CD15/SSEA, CD44, or A2B5 for CSC isolation. However, several authors report that yet no single marker can define a universal GBM CSC population. The identity of GBM CSCs is still unresolved and, although widely used, there is controversy whether marker-expressing cells fulfill the functional criteria of bona fide CSCs (Nat Commun. 2019 Apr 16;10(1):1787. doi: 10.1038/s41467-019-09853-z). In this context, functional assays combined with marker expression are indispensable for the validation of selected CSC subpopulation.
  • Line 170: to support the sentence please report the statistical analysis of TMZ alone vs TMZ+PALM. Please also specify the concentration of palmitate use and when it was applied in the experimental procedure.
  • Line 182: Table 1 is missing
  • 3D, both the statistical analysis and error bars are missing
  • Section 2.4 must be completely revised: several points must be clarified. First, the authors must introduce statistical analysis. Considering the presence of a drug and several time points the authors should use a two-way ANOVA as statistical analysis. The authors should report in supplementary files a representative image of each time point for the GBM43. It is not clear the use of “Total” in the figure, does it refers to CD133lo (differentiated cells)? How do the authors explain the discrepancy of data in the representative plots and graphics in fig. 4 and supplementary figures?
  • Despite the method used to assess glucose and FA uptake could be of interest, the authors should perform other experiments to support their conclusions.
  • Line 240: the authors must provide evidence (from literature or experimental) to state the classification of the GBM line used.
  • Section 2.5: despite the results are discussed, the data on GSC cells are missing.
  • Line 294: did the authors performed an accurate revision of the literature? This sentence must be removed (J Exp Clin Cancer Res. 2015 Oct 6;34:111. doi: 10.1186/s13046-015-0221-y; J Exp Clin Cancer Res. 2019; 38: 218. doi: 10.1186/s13046-019-1214-z).
  • The authors test only the effects of TMZ, to better reflect the reported data, the authors should change the title from “Chemotherapeutic stress alters metabolic behavior in glioblastoma stem cells” in “Temozolomide alters metabolic behavior in glioblastoma stem-like cells”

Author Response

Reviewer 2

In the manuscript entitled “Chemotherapeutic stress alters metabolic behavior in glioblastoma stem cells”, the authors performed a characterization of lipid and glucose uptake in different PDX GBM cells and after the treatment with TMZ (the chemotherapeutic agent used in clinical practice). The authors reported the increase of fatty acid uptake based on TMZ treatment, the GBM cell phenotype, and the amount of stem cell counterpart identified as CD133 + cells. Despite the characterization of the metabolic shift in GBM cells after TMZ treatment could be of interest, the manuscript has several criticisms, and in the present form is not suitable for publication in Cancers journal.

Thank you very much for reviewing our manuscript. We are grateful for your insight and suggestions for improvement. We have made many changes to the draft and endeavored to utilize your suggestions to improve our paper. Please see below for a line-by-line discussion of your main critiques. 

The main criticisms:

1. The statistical analysis of the data is missing throughout the manuscript and in supplementary figures.

Response: We have increased the description of statistics utilized throughout the manuscript.

2. The introduction must be improved, several data on GBM and GSC metabolism has been reported (see the review Front Cell Dev Biol. 2017 Apr 26;5:43. doi: 10.3389/fcell.2017.00043. eCollection 2017; Neuropathology. 2019 Feb;39(1):3-13. doi: 10.1111/neup.12535; Front Oncol. 2019 Mar 6;9:118. doi: 10.3389/fonc.2019.00118. Article: Cell Death Dis 11, 253 (2020). https://doi.org/10.1038/s41419-020-2449-5).

Response: Thank you for providing these specific references. We have included them in our introduction and discussion sections in order to more accurately describe our results and their significance (Lines 60-70):

In light of this complex interplay between environment, phenotype, and tumor progression, understanding the mechanisms that underlie this plasticity is critical. To that end, a range of phenotypic domains have been examined for contributions to tumor plasticity during therapy, including epigenetic modification and gene expression [1,2]. Recently, the metabolic phenotype of GBM tumors has emerged as an exciting area of research (expertly reviewed [3,4]). GBM tumors have been shown to rely on glycolysis during phases of growth [5], with inhibition of glycolysis improving survival of mouse models of GBM[6,7]. Other groups have highlighted the importance of fatty acid oxidation in GBM growth[8]. Critically, metabolism in brain tumors is heterogeneous and linked to stemness. For example, a GSC population has been demonstrated to exhibit elevated lipid levels and utilize fatty acids to withstand nutrient deprivation[9]. Other reports have highlighted the importance of fatty acid synthesis in enabling adoption of a GSC state[10].

3. IDH1 mutation is the main genetic mutation driving GBM development (see the above review). The isocitrate dehydrogenase enzymes are involved in the oxidative carboxylation of isocitrate to α‐ketoglutarate (α‐KG) in the cytosol and mitochondria. Of note, mutant IDH is relevant to glioma pathogenesis through its association with metabolic reprogramming. Based on the manuscript rationale, the authors must report the IDH1/2 status of all the used cell lines.

Response:
We appreciate this very important point. IDH status has been included in the manuscript for each cell line. We have also included a section in the discussion specifically on the importance of IDH status in glioma metabolic plasticity. The following link provides more data on the PDX GBM lines utilized in these experiments (https://www.mayo.edu/research/labs/translational-neuro-oncology/mayo-clinic-brain-tumor-patient-derived-xenograft-national-resource/pdx-genotype/common-genomic-alterations).

Lines 589-602 were added:

This study is not without limitations. First, the technical difficulty of culturing large number of GSCs limited our ability to probe a confirmed post-TMZ sample of GSCs by Seahorse. Second, GBM tumor cells are exquisitely sensitive to changes in culture conditions[16,17]. While we were able to confirm our major finding in a mouse model, there is always concern that findings may not perfectly recapitulate the physiological conditions of GBM tumors in human patients. Third, we only investigated a specific subtype of fatty acids. Lipid metabolism in cancer is an expansive area of research [18-20]; we hope that further exploration will reveal further details about the connection between specific lipid particles and GBM plasticity. Finally, these experiments were conducted in cell lines that are all isocitrate dehydrogenase (IDH1/2) wild type (see methods). IDH status is a key driver of differential prognosis in GBM [21,22]. As such, these data can only be applied to IDH wildtype cell lines and tumors. Further research will establish the role of IDH mutations in GBM therapy-induced plasticity. Despite these limitations, this study highlights the importance of metabolic plasticity in GBM and its potential as a therapeutic target for increasing the efficacy of TMZ.

4. The methods used for the experiments are messy. The author should reorganize the section in several parts reporting for each experiment the time of in vitro culturing, the cell density used, and the media used. All this information is pivotal to interpreter the reported data. 

Response: We apologize the methods were difficult to interpret. We have altered both the methods section and the description of the methods used in the results section.

5. Line 65: please specify the protocol used for CSC enrichment. Furthermore, similar data have been already reported in the literature (Proc Natl Acad Sci U S A. 2011 Sep 20; 108(38): 16062–16067. doi: 10.1073/pnas.1106704108)

Response: We have expanded our description of this protocol in both the results and methods section. Thank you for bringing this 2011 reference to our attention. We have added it to our discussion. Lines 110-115:

To begin our investigation into the connection between cellular phenotype and metabolic phenotype, we first compared metabolic activity of patient derived xenografts (PDX) GBM 43 cells in culture conditions known to induce different cellular phenotypes. GBM cells are known to be highly sensitive to culture conditions[17,23]. GBM43 cells were cultured either as neurospheres in media that promotes a GSC fate (Neurobasal supplemented with N2, bFGF, EGF [24]) or compared to adherent in conditions known to force differentiation (10% fetal bovine serum) [25].

6. The reference to Fig. S1 and Fig S2 are missing.

Response: Please see section 2.2. We had previously listed them as Supp. Figures 1 and 2, which we have now changed to Figures S1 and S2 to more closely align with formatting guidelines.

7. The authors used the CD133 as a marker to select the CSC subpopulation. In GBM many studies rely on cell surface markers such as CD133, CD15/SSEA, CD44, or A2B5 for CSC isolation. However, several authors report that yet no single marker can define a universal GBM CSC population. The identity of GBM CSCs is still unresolved and, although widely used, there is controversy whether marker-expressing cells fulfill the functional criteria of bona fide CSCs (Nat Commun. 2019 Apr 16;10(1):1787. doi: 10.1038/s41467-019-09853-z). In this context, functional assays combined with marker expression are indispensable for the validation of selected CSC subpopulation.

Response: This point is key to all research on GBM and we appreciate the reviewer raising it. We have extensively investigated the issue pretending to the marker that can faithfully identify CSC subpopulation during chemotherapy in our previous publications (PMID:2460879: Supplementary Fig. 1;  27765847: Fig. 2, Supplementary Fig. 2, 34) demonstrated that at least in our PDX model CD133 can function as CSC markers during TMZ therapy. For example, In vivo limiting dilution assay using intracranially implanted GBM43 xenografts (100 and 500 cells/animal) sorted for CD133 only, CD15 only, and double negative cells (CD133-CD15-). Kaplan-Meyer survival plot shows differences in tumor engraftment show that CD133 positive cells with significantly higher tumor engraftment capacity as compared to double negative or CD15 positive cells. We have developed multiple CSC-specific reporter systems and have demonstrated that CD133 positive cells in our PDX models also express elevated level other CSC-specific transcription factors as well as phenotypic markers both in vitro and in vivo (27765847: Fig. 2).

8. Line 170: to support the sentence please report the statistical analysis of TMZ alone vs TMZ+PALM. Please also specify the concentration of palmitate use and when it was applied in the experimental procedure.

Response: The concentration of palmitate used for every assay was 150µm. This is based off of the following technical brief “Simultaneously Measuring Oxidation of Exogenous and Endogenous Fatty Acids using the XF Palmitate-BSA FAO Substrate with the XF Cell Mito Stress Test”. For Seahorse Analysis, Palmitate-BSA was added immediately prior to loading the plate into the analyzer for Seahorse analysis.

Statistics for this experiment were described in the figure legend (Figure 3). These groups were compared using One-Way ANOVA with Tukey’s Post-Hoc. The PALM concentration has been added throughout the manuscript.

9. Line 182: Table 1 is missing

Response: We regret this omission. It has been added to the supplementary materials file.

10. 3D, both the statistical analysis and error bars are missing

Response: We apologize for this oversight. We have removed that portion of the figure and instead focused the reader’s attention on the microarray data, which was robust and statistically significant.

11.Section 2.4 must be completely revised: several points must be clarified. First, the authors must introduce statistical analysis. Considering the presence of a drug and several time points the authors should use a two-way ANOVA as statistical analysis. The authors should report in supplementary files a representative image of each time point for the GBM43. It is not clear the use of “Total” in the figure, does it refers to CD133lo (differentiated cells)? How do the authors explain the discrepancy of data in the representative plots and graphics in fig. 4 and supplementary figures?

Response: Section 2.4 has been revised to better detail statistics utilized.

We apologize for the confusion around “Total”. This refers to bulk tumor sample assessment, without selective gating for either GSCs or differentiated GBM cells.

We did not identify the discrepancy between plots and graphics in figure 4 / the supplementary figures. Could you please clarify the discrepancy you are referencing?

12. Despite the method used to assess glucose and FA uptake could be of interest, the authors should perform other experiments to support their conclusions.

Response: We thank the reviewer for this thoughtful comment. As we have discussed, assessment of glucose and FA uptake in the small GSC population presents a number of technical challenging. It required the in-house production of a purpose-built reagent [26]. This work is time and labor intensive. These challenges are even more pronounced with in vivo assessments, which were able to generate. In addition to that unique methodology, we were able to corroborate our results with metabolic assays considered standard in the field of metabolism. While we agree that more methods can always improve manuscripts, we would like to respectfully posit that these results – encompassing animal models, PDX cells lines, metabolic assessments, and transcriptomics – represents a strong statement on the importance of metabolic plasticity during chemotherapy in GBM.

13. Line 240: the authors must provide evidence (from literature or experimental) to state the classification of the GBM line used.

Response: We have added citations on the lines of GBM cells that we utilize in our lab and their subtyping. Thank you for bringing this omission to our attention.

14. Section 2.5: despite the results are discussed, the data on GSC cells are missing.

Response: Please see the updated Figure 6 and section 2.5 for further discussion of the GSC population in vivo. We apologize for the oversight in figure finalization.

15. Line 294: did the authors performed an accurate revision of the literature? This sentence must be removed (J Exp Clin Cancer Res. 2015 Oct 6;34:111. doi: 10.1186/s13046-015-0221-y; J Exp Clin Cancer Res. 2019; 38: 218. doi: 10.1186/s13046-019-1214-z).

Response: Thank you for providing these references. We apologize for the previous phrasing, which was inaccurate. This line has been removed.

16. The authors test only the effects of TMZ, to better reflect the reported data, the authors should change the title from “Chemotherapeutic stress alters metabolic behavior in glioblastoma stem cells” in “Temozolomide alters metabolic behavior in glioblastoma stem-like cells”

Response: We have updated to the title to “Temozolomide treatment increases fatty acid uptake in glioblastoma stem cells” in order to be more specific about our manuscript, as requested by this reviewer and reviewer number 1.

Please find attached our revised manuscript where all the changes are highlighted yellow.

REFERENCES

  1. Liau, B.B.; Sievers, C.; Donohue, L.K.; Gillespie, S.M.; Flavahan, W.A.; Miller, T.E.; Venteicher, A.S.; Hebert, C.H.; Carey, C.D.; Rodig, S.J., et al. Adaptive Chromatin Remodeling Drives Glioblastoma Stem Cell Plasticity and Drug Tolerance. Cell Stem Cell 2017, 20, 233-246 e237, doi:10.1016/j.stem.2016.11.003.
  2. Heddleston, J.M.; Li, Z.; McLendon, R.E.; Hjelmeland, A.B.; Rich, J.N. The hypoxic microenvironment maintains glioblastoma stem cells and promotes reprogramming towards a cancer stem cell phenotype. Cell Cycle 2009, 8, 3274-3284, doi:10.4161/cc.8.20.9701.
  3. Badr, C.E.; Silver, D.J.; Siebzehnrubl, F.A.; Deleyrolle, L.P. Metabolic heterogeneity and adaptability in brain tumors. Cell Mol Life Sci 2020, 10.1007/s00018-020-03569-w, doi:10.1007/s00018-020-03569-w.
  4. Masui, K.; Onizuka, H.; Cavenee, W.K.; Mischel, P.S.; Shibata, N. Metabolic reprogramming in the pathogenesis of glioma: Update. Neuropathology 2019, 39, 3-13, doi:10.1111/neup.12535.
  5. Marin-Valencia, I.; Yang, C.; Mashimo, T.; Cho, S.; Baek, H.; Yang, X.L.; Rajagopalan, K.N.; Maddie, M.; Vemireddy, V.; Zhao, Z., et al. Analysis of tumor metabolism reveals mitochondrial glucose oxidation in genetically diverse human glioblastomas in the mouse brain in vivo. Cell Metab 2012, 15, 827-837, doi:10.1016/j.cmet.2012.05.001.
  6. Tateishi, K.; Iafrate, A.J.; Ho, Q.; Curry, W.T.; Batchelor, T.T.; Flaherty, K.T.; Onozato, M.L.; Lelic, N.; Sundaram, S.; Cahill, D.P., et al. Myc-Driven Glycolysis Is a Therapeutic Target in Glioblastoma. Clin Cancer Res 2016, 22, 4452-4465, doi:10.1158/1078-0432.CCR-15-2274.
  7. Sanzey, M.; Abdul Rahim, S.A.; Oudin, A.; Dirkse, A.; Kaoma, T.; Vallar, L.; Herold-Mende, C.; Bjerkvig, R.; Golebiewska, A.; Niclou, S.P. Comprehensive analysis of glycolytic enzymes as therapeutic targets in the treatment of glioblastoma. PLoS One 2015, 10, e0123544, doi:10.1371/journal.pone.0123544.
  1. Lin, H.; Patel, S.; Affleck, V.S.; Wilson, I.; Turnbull, D.M.; Joshi, A.R.; Maxwell, R.; Stoll, E.A. Fatty acid oxidation is required for the respiration and proliferation of malignant glioma cells. Neuro Oncol 2017, 19, 43-54, doi:10.1093/neuonc/now128.
  2. Hoang-Minh, L.B.; Siebzehnrubl, F.A.; Yang, C.; Suzuki-Hatano, S.; Dajac, K.; Loche, T.; Andrews, N.; Schmoll Massari, M.; Patel, J.; Amin, K., et al. Infiltrative and drug-resistant slow-cycling cells support metabolic heterogeneity in glioblastoma. EMBO J 2018, 37, doi:10.15252/embj.201798772.
  3. Yasumoto, Y.; Miyazaki, H.; Vaidyan, L.K.; Kagawa, Y.; Ebrahimi, M.; Yamamoto, Y.; Ogata, M.; Katsuyama, Y.; Sadahiro, H.; Suzuki, M., et al. Inhibition of Fatty Acid Synthase Decreases Expression of Stemness Markers in Glioma Stem Cells. PLoS One 2016, 11, e0147717, doi:10.1371/journal.pone.0147717.
  4. Zhang, J.; Zhang, Q. Using Seahorse Machine to Measure OCR and ECAR in Cancer Cells. Methods Mol Biol 2019, 1928, 353-363, doi:10.1007/978-1-4939-9027-6_18.
  5. Yepez, V.A.; Kremer, L.S.; Iuso, A.; Gusic, M.; Kopajtich, R.; Konarikova, E.; Nadel, A.; Wachutka, L.; Prokisch, H.; Gagneur, J. OCR-Stats: Robust estimation and statistical testing of mitochondrial respiration activities using Seahorse XF Analyzer. PLoS One 2018, 13, e0199938, doi:10.1371/journal.pone.0199938.
  6. Giannini, C.; Sarkaria, J.N.; Saito, A.; Uhm, J.H.; Galanis, E.; Carlson, B.L.; Schroeder, M.A.; James, C.D. Patient tumor EGFR and PDGFRA gene amplifications retained in an invasive intracranial xenograft model of glioblastoma multiforme. Neuro-oncology 2005, 7, 164-176.
  7. Verhaak, R.G.; Hoadley, K.A.; Purdom, E.; Wang, V.; Qi, Y.; Wilkerson, M.D.; Miller, C.R.; Ding, L.; Golub, T.; Mesirov, J.P., et al. Integrated genomic analysis identifies clinically relevant subtypes of glioblastoma characterized by abnormalities in PDGFRA, IDH1, EGFR, and NF1. Cancer Cell 2010, 17, 98-110, doi:10.1016/j.ccr.2009.12.020.
  8. Thakkar, J.P.; Dolecek, T.A.; Horbinski, C.; Ostrom, Q.T.; Lightner, D.D.; Barnholtz-Sloan, J.S.; Villano, J.L. Epidemiologic and molecular prognostic review of glioblastoma. Cancer Epidemiol Biomarkers Prev 2014, 23, 1985-1996, doi:10.1158/1055-9965.EPI-14-0275.
  9. Strickland, M.; Stoll, E.A. Metabolic Reprogramming in Glioma. Front Cell Dev Biol 2017, 5, 43, doi:10.3389/fcell.2017.00043.
  10. Caragher, S.; Chalmers, A.J.; Gomez-Roman, N. Glioblastoma's Next Top Model: Novel Culture Systems for Brain Cancer Radiotherapy Research. Cancers (Basel) 2019, 11, doi:10.3390/cancers11010044.
  11. Carracedo, A.; Cantley, L.C.; Pandolfi, P.P. Cancer metabolism: fatty acid oxidation in the limelight. Nat Rev Cancer 2013, 13, 227-232, doi:10.1038/nrc3483.
  12. Rohrig, F.; Schulze, A. The multifaceted roles of fatty acid synthesis in cancer. Nat Rev Cancer 2016, 16, 732-749, doi:10.1038/nrc.2016.89.
  13. Griffin, J.L.; Shockcor, J.P. Metabolic profiles of cancer cells. Nat Rev Cancer 2004, 4, 551-561, doi:10.1038/nrc1390.
  14. Huse, J.T.; Holland, E.C. Targeting brain cancer: advances in the molecular pathology of malignant glioma and medulloblastoma. Nat Rev Cancer 2010, 10, 319-331, doi:10.1038/nrc2818.
  15. SongTao, Q.; Lei, Y.; Si, G.; YanQing, D.; HuiXia, H.; XueLin, Z.; LanXiao, W.; Fei, Y. IDH mutations predict longer survival and response to temozolomide in secondary glioblastoma. Cancer Sci 2012, 103, 269-273, doi:10.1111/j.1349-7006.2011.02134.x.
  16. Garnier, D.; Renoult, O.; Alves-Guerra, M.C.; Paris, F.; Pecqueur, C. Glioblastoma Stem-Like Cells, Metabolic Strategy to Kill a Challenging Target. Front Oncol 2019, 9, 118, doi:10.3389/fonc.2019.00118.
  17. Podergajs, N.; Brekka, N.; Radlwimmer, B.; Herold-Mende, C.; Talasila, K.M.; Tiemann, K.; Rajcevic, U.; Lah, T.T.; Bjerkvig, R.; Miletic, H. Expansive growth of two glioblastoma stem-like cell lines is mediated by bFGF and not by EGF. Radiol Oncol 2013, 47, 330-337, doi:10.2478/raon-2013-0063.
  18. Stockhausen, M.T.; Kristoffersen, K.; Stobbe, L.; Poulsen, H.S. Differentiation of glioblastoma multiforme stem-like cells leads to downregulation of EGFR and EGFRvIII and decreased tumorigenic and stem-like cell potential. Cancer Biol Ther 2014, 15, 216-224, doi:10.4161/cbt.26736.
  19. Muroski, M.E.; Miska, J.; Chang, A.L.; Zhang, P.; Rashidi, A.; Moore, H.; Lopez-Rosas, A.; Han, Y.; Lesniak, M.S. Fatty Acid Uptake in T Cell Subsets Using a Quantum Dot Fatty Acid Conjugate. Sci Rep 2017, 7, 5790, doi:10.1038/s41598-017-05556-x.
  20. Wang, D.; Green, M.F.; McDonnell, E.; Hirschey, M.D. Oxygen flux analysis to understand the biological function of sirtuins. Methods Mol Biol 2013, 1077, 241-258, doi:10.1007/978-1-62703-637-5_16.
  21. van der Windt, G.J.W.; Chang, C.H.; Pearce, E.L. Measuring Bioenergetics in T Cells Using a Seahorse Extracellular Flux Analyzer. Curr Protoc Immunol 2016, 113, 3 16B 11-13 16B 14, doi:10.1002/0471142735.im0316bs113.
  22. Nicholas, D.; Proctor, E.A.; Raval, F.M.; Ip, B.C.; Habib, C.; Ritou, E.; Grammatopoulos, T.N.; Steenkamp, D.; Dooms, H.; Apovian, C.M., et al. Advances in the quantification of mitochondrial function in primary human immune cells through extracellular flux analysis. PLoS One 2017, 12, e0170975, doi:10.1371/journal.pone.0170975.
  23. Nicholls, D.G.; Darley-Usmar, V.M.; Wu, M.; Jensen, P.B.; Rogers, G.W.; Ferrick, D.A. Bioenergetic profile experiment using C2C12 myoblast cells. J Vis Exp 2010, 10.3791/2511, doi:10.3791/2511.
  24. Huynh, F.K.; Green, M.F.; Koves, T.R.; Hirschey, M.D. Measurement of fatty acid oxidation rates in animal tissues and cell lines. Methods Enzymol 2014, 542, 391-405, doi:10.1016/B978-0-12-416618-9.00020-0.

Reviewer 3 Report

Major Comments:

  1. What metabolite inhibitors were used for the study? Was it an array of metabolite inhibitor or was there any rationale for choosing particular inhibitors in Fig 1?
  2. In Fig 2, only palmitate is used to generate Fatty acid uptake. Why?
  3. Do cells need to be incubated with Eto for some time before palmitate is added into the media at the same time? Please clarify.
  4. In Fig 3, cells treated with TMZ+Palimtatae further enhanced metabolic shift. To fully understand, whether Fatty acid leads to increased mitochondrial metabolism, we need Eto+Palmitate as an experimental parameter to bring down mitochondrial metabolism to normal levels.
  5. Last statement under section 2.3, says TMZ moves metabolism towards FAO. But until now ETO or CPT1 only shows Fatty acid uptake into cells and mitochondria. No where oxidation is mentioned.

Author Response

Reviewer 3

1. What metabolite inhibitors were used for the study? Was it an array of metabolite inhibitor or was there any rationale for choosing particular inhibitors in Fig 1?

Response: We are grateful to you for raising this clarification. The Mitochondrial stress test is a well-accepted and standard protocol for metabolic measurement in cells. There are well over 1000 publications on this methodology. See link below:

https://www.agilent.com/en/product/cell-analysis/real-time-cell-metabolic-analysis/xf-assay-kits-reagents-cell-assay-media/seahorse-xf-cell-mito-stress-test-kit-740885

Regarding its use in Fatty acid metabolic measurements, please refer to the following technical brief: “Simultaneously Measuring Oxidation of Exogenous and Endogenous Fatty Acids using the XF Palmitate-BSA FAO Substrate with the XF Cell Mito Stress Test”. This methodology was based off of methodology published by Wang et al [27].

If you are interested in further context on these experiments, please see the following manuscripts[12,28-30].

2. In Fig 2, only palmitate is used to generate Fatty acid uptake. Why?

Response: Palmitate oxidation is a robust methodology for assessing fatty acid oxidation in cells. As it is wholly saturated fatty acid, it can be rapidly oxidized to CO2 [31]. A useful alternative is oleic acid:BSA which we did not perform. Our goal was to assess the effects of lipid uptake/oxidation by therapeutic stress, not specify which lipid species are being used. This is the subject of our future work, and we have provided this discussion as a limitation of our study. The text we added in below:

“Third, we only investigated a specific subtype of fatty acids. Lipid metabolism is an expansive area of research; we hope that further exploration will reveal further details about the connection between specific lipid particles and GBM plasticity.”

3. Do cells need to be incubated with Eto for some time before palmitate is added into the media at the same time? Please clarify.

Response: We apologize for the lack of clarity on experimental design. Etomoxir was added 30 minutes before addition of palmitate as suggested by the technical brief above. We have included this technical information in the methods.

4. In Fig 3, cells treated with TMZ+Palimtatae further enhanced metabolic shift. To fully understand, whether Fatty acid leads to increased mitochondrial metabolism, we need Eto+Palmitate as an experimental parameter to bring down mitochondrial metabolism to normal levels. 

Response: We have performed this assay and included the data in Figure 3. The data shows that palmitate administration does not further increase fatty acid oxidation beyond TMZ administration.

5. Last statement under section 2.3, says TMZ moves metabolism towards FAO. But until now ETO or CPT1 only shows Fatty acid uptake into cells and mitochondria. No where oxidation is mentioned.

Response: This is a fair concern. Thank you for bringing it to our attention. We have changed the text to more appropriately reflect the data:

“In sum, these results show that TMZ treatment increases the avidity of GBM cells for fatty acids.”

REFERENCES

  1. Liau, B.B.; Sievers, C.; Donohue, L.K.; Gillespie, S.M.; Flavahan, W.A.; Miller, T.E.; Venteicher, A.S.; Hebert, C.H.; Carey, C.D.; Rodig, S.J., et al. Adaptive Chromatin Remodeling Drives Glioblastoma Stem Cell Plasticity and Drug Tolerance. Cell Stem Cell 2017, 20, 233-246 e237, doi:10.1016/j.stem.2016.11.003.
  2. Heddleston, J.M.; Li, Z.; McLendon, R.E.; Hjelmeland, A.B.; Rich, J.N. The hypoxic microenvironment maintains glioblastoma stem cells and promotes reprogramming towards a cancer stem cell phenotype. Cell Cycle 2009, 8, 3274-3284, doi:10.4161/cc.8.20.9701.
  3. Badr, C.E.; Silver, D.J.; Siebzehnrubl, F.A.; Deleyrolle, L.P. Metabolic heterogeneity and adaptability in brain tumors. Cell Mol Life Sci 2020, 10.1007/s00018-020-03569-w, doi:10.1007/s00018-020-03569-w.
  4. Masui, K.; Onizuka, H.; Cavenee, W.K.; Mischel, P.S.; Shibata, N. Metabolic reprogramming in the pathogenesis of glioma: Update. Neuropathology 2019, 39, 3-13, doi:10.1111/neup.12535.
  5. Marin-Valencia, I.; Yang, C.; Mashimo, T.; Cho, S.; Baek, H.; Yang, X.L.; Rajagopalan, K.N.; Maddie, M.; Vemireddy, V.; Zhao, Z., et al. Analysis of tumor metabolism reveals mitochondrial glucose oxidation in genetically diverse human glioblastomas in the mouse brain in vivo. Cell Metab 2012, 15, 827-837, doi:10.1016/j.cmet.2012.05.001.
  6. Tateishi, K.; Iafrate, A.J.; Ho, Q.; Curry, W.T.; Batchelor, T.T.; Flaherty, K.T.; Onozato, M.L.; Lelic, N.; Sundaram, S.; Cahill, D.P., et al. Myc-Driven Glycolysis Is a Therapeutic Target in Glioblastoma. Clin Cancer Res 2016, 22, 4452-4465, doi:10.1158/1078-0432.CCR-15-2274.
  7. Sanzey, M.; Abdul Rahim, S.A.; Oudin, A.; Dirkse, A.; Kaoma, T.; Vallar, L.; Herold-Mende, C.; Bjerkvig, R.; Golebiewska, A.; Niclou, S.P. Comprehensive analysis of glycolytic enzymes as therapeutic targets in the treatment of glioblastoma. PLoS One 2015, 10, e0123544, doi:10.1371/journal.pone.0123544.
  1. Lin, H.; Patel, S.; Affleck, V.S.; Wilson, I.; Turnbull, D.M.; Joshi, A.R.; Maxwell, R.; Stoll, E.A. Fatty acid oxidation is required for the respiration and proliferation of malignant glioma cells. Neuro Oncol 2017, 19, 43-54, doi:10.1093/neuonc/now128.
  2. Hoang-Minh, L.B.; Siebzehnrubl, F.A.; Yang, C.; Suzuki-Hatano, S.; Dajac, K.; Loche, T.; Andrews, N.; Schmoll Massari, M.; Patel, J.; Amin, K., et al. Infiltrative and drug-resistant slow-cycling cells support metabolic heterogeneity in glioblastoma. EMBO J 2018, 37, doi:10.15252/embj.201798772.
  3. Yasumoto, Y.; Miyazaki, H.; Vaidyan, L.K.; Kagawa, Y.; Ebrahimi, M.; Yamamoto, Y.; Ogata, M.; Katsuyama, Y.; Sadahiro, H.; Suzuki, M., et al. Inhibition of Fatty Acid Synthase Decreases Expression of Stemness Markers in Glioma Stem Cells. PLoS One 2016, 11, e0147717, doi:10.1371/journal.pone.0147717.
  4. Zhang, J.; Zhang, Q. Using Seahorse Machine to Measure OCR and ECAR in Cancer Cells. Methods Mol Biol 2019, 1928, 353-363, doi:10.1007/978-1-4939-9027-6_18.
  5. Yepez, V.A.; Kremer, L.S.; Iuso, A.; Gusic, M.; Kopajtich, R.; Konarikova, E.; Nadel, A.; Wachutka, L.; Prokisch, H.; Gagneur, J. OCR-Stats: Robust estimation and statistical testing of mitochondrial respiration activities using Seahorse XF Analyzer. PLoS One 2018, 13, e0199938, doi:10.1371/journal.pone.0199938.
  6. Giannini, C.; Sarkaria, J.N.; Saito, A.; Uhm, J.H.; Galanis, E.; Carlson, B.L.; Schroeder, M.A.; James, C.D. Patient tumor EGFR and PDGFRA gene amplifications retained in an invasive intracranial xenograft model of glioblastoma multiforme. Neuro-oncology 2005, 7, 164-176.
  7. Verhaak, R.G.; Hoadley, K.A.; Purdom, E.; Wang, V.; Qi, Y.; Wilkerson, M.D.; Miller, C.R.; Ding, L.; Golub, T.; Mesirov, J.P., et al. Integrated genomic analysis identifies clinically relevant subtypes of glioblastoma characterized by abnormalities in PDGFRA, IDH1, EGFR, and NF1. Cancer Cell 2010, 17, 98-110, doi:10.1016/j.ccr.2009.12.020.
  8. Thakkar, J.P.; Dolecek, T.A.; Horbinski, C.; Ostrom, Q.T.; Lightner, D.D.; Barnholtz-Sloan, J.S.; Villano, J.L. Epidemiologic and molecular prognostic review of glioblastoma. Cancer Epidemiol Biomarkers Prev 2014, 23, 1985-1996, doi:10.1158/1055-9965.EPI-14-0275.
  9. Strickland, M.; Stoll, E.A. Metabolic Reprogramming in Glioma. Front Cell Dev Biol 2017, 5, 43, doi:10.3389/fcell.2017.00043.
  10. Caragher, S.; Chalmers, A.J.; Gomez-Roman, N. Glioblastoma's Next Top Model: Novel Culture Systems for Brain Cancer Radiotherapy Research. Cancers (Basel) 2019, 11, doi:10.3390/cancers11010044.
  11. Carracedo, A.; Cantley, L.C.; Pandolfi, P.P. Cancer metabolism: fatty acid oxidation in the limelight. Nat Rev Cancer 2013, 13, 227-232, doi:10.1038/nrc3483.
  12. Rohrig, F.; Schulze, A. The multifaceted roles of fatty acid synthesis in cancer. Nat Rev Cancer 2016, 16, 732-749, doi:10.1038/nrc.2016.89.
  13. Griffin, J.L.; Shockcor, J.P. Metabolic profiles of cancer cells. Nat Rev Cancer 2004, 4, 551-561, doi:10.1038/nrc1390.
  14. Huse, J.T.; Holland, E.C. Targeting brain cancer: advances in the molecular pathology of malignant glioma and medulloblastoma. Nat Rev Cancer 2010, 10, 319-331, doi:10.1038/nrc2818.
  15. SongTao, Q.; Lei, Y.; Si, G.; YanQing, D.; HuiXia, H.; XueLin, Z.; LanXiao, W.; Fei, Y. IDH mutations predict longer survival and response to temozolomide in secondary glioblastoma. Cancer Sci 2012, 103, 269-273, doi:10.1111/j.1349-7006.2011.02134.x.
  16. Garnier, D.; Renoult, O.; Alves-Guerra, M.C.; Paris, F.; Pecqueur, C. Glioblastoma Stem-Like Cells, Metabolic Strategy to Kill a Challenging Target. Front Oncol 2019, 9, 118, doi:10.3389/fonc.2019.00118.
  17. Podergajs, N.; Brekka, N.; Radlwimmer, B.; Herold-Mende, C.; Talasila, K.M.; Tiemann, K.; Rajcevic, U.; Lah, T.T.; Bjerkvig, R.; Miletic, H. Expansive growth of two glioblastoma stem-like cell lines is mediated by bFGF and not by EGF. Radiol Oncol 2013, 47, 330-337, doi:10.2478/raon-2013-0063.
  18. Stockhausen, M.T.; Kristoffersen, K.; Stobbe, L.; Poulsen, H.S. Differentiation of glioblastoma multiforme stem-like cells leads to downregulation of EGFR and EGFRvIII and decreased tumorigenic and stem-like cell potential. Cancer Biol Ther 2014, 15, 216-224, doi:10.4161/cbt.26736.
  19. Muroski, M.E.; Miska, J.; Chang, A.L.; Zhang, P.; Rashidi, A.; Moore, H.; Lopez-Rosas, A.; Han, Y.; Lesniak, M.S. Fatty Acid Uptake in T Cell Subsets Using a Quantum Dot Fatty Acid Conjugate. Sci Rep 2017, 7, 5790, doi:10.1038/s41598-017-05556-x.
  20. Wang, D.; Green, M.F.; McDonnell, E.; Hirschey, M.D. Oxygen flux analysis to understand the biological function of sirtuins. Methods Mol Biol 2013, 1077, 241-258, doi:10.1007/978-1-62703-637-5_16.
  21. van der Windt, G.J.W.; Chang, C.H.; Pearce, E.L. Measuring Bioenergetics in T Cells Using a Seahorse Extracellular Flux Analyzer. Curr Protoc Immunol 2016, 113, 3 16B 11-13 16B 14, doi:10.1002/0471142735.im0316bs113.
  22. Nicholas, D.; Proctor, E.A.; Raval, F.M.; Ip, B.C.; Habib, C.; Ritou, E.; Grammatopoulos, T.N.; Steenkamp, D.; Dooms, H.; Apovian, C.M., et al. Advances in the quantification of mitochondrial function in primary human immune cells through extracellular flux analysis. PLoS One 2017, 12, e0170975, doi:10.1371/journal.pone.0170975.
  23. Nicholls, D.G.; Darley-Usmar, V.M.; Wu, M.; Jensen, P.B.; Rogers, G.W.; Ferrick, D.A. Bioenergetic profile experiment using C2C12 myoblast cells. J Vis Exp 2010, 10.3791/2511, doi:10.3791/2511.
  24. Huynh, F.K.; Green, M.F.; Koves, T.R.; Hirschey, M.D. Measurement of fatty acid oxidation rates in animal tissues and cell lines. Methods Enzymol 2014, 542, 391-405, doi:10.1016/B978-0-12-416618-9.00020-0.

Please find attached our revised manuscript where all the changes are highlighted yellow.
